



# **Tributaries affect the thermal response of lakes to climate**
# **change**
Love Råman Vinnå[1], Alfred Wüest[1,2], Massimiliano Zappa[3], Gabriel Fink[4], Damien
Bouffard[1,2]
[1]Aquatic Systems Laboratory, Margaretha Kamprad Chair, École Polytechnique Fédérale de Lausanne, Institute
of Environmental Engineering, Lausanne CH-1015, Switzerland.
[2]Eawag, Swiss Federal Institute of Aquatic Science and Technology, Surface Waters - Research and
Management, Kastanienbaum, Switzerland.
[3]Swiss Federal Institute for Forest, Snow and Landscape Research
WSL, Birmensdorf, Switzerland.
[4]Center for Environmental Systems Research, CESR, University of Kassel, Kassel,
*Correspondence to:* Love Råman Vinnå (love.ramanvinna@epfl.ch)





**Abstract**
Thermal responses of inland waters to climate change varies on global and regional scales. The extent of warming
is determined by system-specific characteristics such as fluvial input. Here we examine the impact of ongoing
climate change on two alpine tributaries, the Aare River and Rhône River, and their respective downstream, peri-
alpine lakes: Lake Biel and Lake Geneva. We propagate regional atmospheric temperature effects into river
discharge projections. These, together with anthropogenic heat sources, are in turn incorporated into simple and
efficient deterministic models that predict future water temperatures, river-borne suspended sediment
concentration (SSC), lake stratification and river intrusion depth/volume into the lakes. Climate-induced shifts in
river discharge regimes, including seasonal flow variations, act as positive and negative feedbacks in influencing
river water temperature and SSC. Discharge-dependent increase of river temperature in turn results in large
seasonal shifts in warming of downstream lakes. Their hydraulic residence times control seasonal variations in
climate-induced heating. Previous studies suggest that climate change will diminish deep-water oxygen renewal
in lakes. We find that climate-related seasonal variations in river temperatures and SSC affect the rate of deep-
penetrating river intrusions. Seasonal variations (with decreasing flows in summer and increasing flows in winter)
determine water input reaching the deepest parts of lakes. The river-lake model results described here show an
annual increase in deep-water renewal. This process may therefore counteract otherwise negative effects of
climate change on deep-water oxygen supply in lakes. Our findings provide a template for evaluating the response
of similar hydrologic systems to on-going climate change.
**Copyright statement**





## 1 Introduction

The thermal and hydrodynamic responses of lakes to climate change are considerably diverse. Observed responses vary on global, regional and even local scales (O'Reilly et al., 2015). Even neighbouring freshwater bodies can react differently to a given increase in air temperature. This indicates that lake-specific characteristics will determine the response to climate change (for clarity and brevity, we refer to anthropogenic climate change simply as 'climate change' or 'climate' from now on). Local factors which affect climate warming of lakes include, among others, morphology (Toffolon et al., 2014), irradiance absorption (Kirillin, 2010; Williamson et al., 2015), local weather conditions (Zhong et al., 2016), stratification (Piccolroaz et al., 2015), atmospheric brightening (Fink et al., 2014a) and ice cover (Austin and Colman, 2007).

Throughflows affect epilimnion and hypolimnion temperatures of lakes. Studies of climate impact typically do not address these sorts of subtleties in lake dynamics due to lack of data or difficulties in predicting future temperature and discharge conditions (Fenocchi et al., 2017). Several studies of large lakes suggest that major tributaries play only a minor role in climate-induced warming and deep-water oxygen renewal (Fink et al., 2014a; Schwefel et al., 2016). Medium- and smaller-scale lakes are, however, more abundant than large lakes (Downing et al., 2006) and exhibit a greater degree of sensitivity to point sources of anthropogenic thermal input which can affect temperature and stratification (Kirillin et al., 2013; Råman Vinnå et al., 2017). Medium- and small-sized lakes also make a more significant contribution to the temperature-dependent global greenhouse gas budget (Holgerson and Raymond, 2016). Accurate prediction of climate change impact therefore requires a more detailed understanding of small- to medium-scale lake and tributary systems.

Climate change exerts dual influence on alpine rivers by introducing variation to both flow and temperatures. Discharge variation takes the form of less flow in summer and more flow in winter due to warmer high altitude snow and ice melt/runoff regimes (Addor et al., 2014; Birsan et al., 2005) which also influence river temperature (Isaak et al., 2012; Van Vliet et al., 2013). Increased air temperature may also enhance erosion rates in river basins thereby supplementing river-borne suspended sediment loads (Bennett et al., 2013). River temperature and suspended sediment content determine water density and by extension, the depth of river plume intrusions into downstream lakes or reservoirs. The depth and volume of river intrusion plumes determine deep-water oxygen renewal, especially for deeper lakes where climate-related warming can reduced seasonal deep convective mixing and thereby deplete deep-water oxygen (Schwefel et al., 2016). Major (deeply penetrating) river intrusion events typically occur due to flooding, which introduces large sediment loads into the river (Fink et al., 2016). The frequency and volume of floods in the Alps are notoriously hard to predict, although a decrease in floods has occurred in association with recent warmer summers observed in the Alps (CH2011, 2011; Glur et al., 2013).

Recent model studies have identified inland waters as risk-hotspots under expected climate change scenarios (Pachauri et al., 2015). These systems require a more detailed analysis given their role in supporting fisheries, agriculture, drinking water production, heat extraction/release and hydropower. This paper examines complex interactions between tributaries and lakes in response to temperature increases and other modifications expected from climate change. Our objectives were to quantify the impact of specific climate change scenarios on (i) alpine tributaries and (ii) downstream peri-alpine lakes with a focus on river-borne suspended sediment concentration (SSC), water temperature, stratification and river intrusion depth and volume.



Our analysis used coupled river-lake models to build on previous research by Fink et al. (2016). These authors
investigated the effect of flood frequencies on deep-water renewal under established climate change scenarios.
Their work did not generate tightly constrained estimates for flooding events. Our analysis therefore provisionally
assumed that flood frequency does not change in the future. In addition to these sources of natural variation, our
models addressed variation in river discharge regimes (i.e. daily mean level shift) under the specified A1B climate
change scenario. These in turn affect SSC and thermal regimes for rivers and their associated downstream lakes.
Our analysis furthermore shows that local point sources of anthropogenic thermal pollution can have major impact
on the response of inland waters to climate change as previously suggested by Fink et al. (2014b).

**2 Methods**
**2.1 Approach**
Investigation of tributary-influence on lake response to climate change followed these procedural steps:
(i)    Define river temperature and SSC models for two major alpine rivers and designate a one-
88          dimensional lake model for a large- and a medium-sized peri-alpine lake.

(ii)   Integrate model (i) with a state-of-the-art river intrusion scheme. Figure 1 shows integration of one-
90          way component models.

(iii)  Obtain and apply estimates of future regional air temperature, tributary discharge and changes in
92          local anthropogenic thermal emissions to both river and lake models.

(iv)   Identify patterns in model outputs of water temperature, SSC, lake stratification and river intrusion
94          parameters (volume and depth).


**2.2 Study area**
This study examined two warm, monomictic, freshwater peri-alpine lakes in western Switzerland, Lake Biel (LB;
7°10' E, 47°5' N) and Lake Geneva (LG; 6°31' E, 46°27' N). Large tributaries originating in the Alps, the Aare
River and Rhône River, feed into LB and LG, respectively (Fig. 2).
LG is a large, meso-eutrophic lake resting at 372 m elevation and covering an area of 580 km$^2$. It reaches a
maximum depth of 309 m and holds a volume of 89 km$^3$ with an average hydraulic residence time of 11.5 years.
Complete seasonal deep convective mixing only occurs on average every fifth winter but is predicted to become
less frequent with on-going climate change (Perroud and Goyette, 2010; Schwefel et al., 2016). The average
surface water temperature has increased by ~0.11° C decade$^{-1}$ to ~0.52° C decade$^{-1}$ (Gillet and QueTin, 2006;
O'Reilly et al., 2015). The Rhône River supplies ~75% of LG's inflow and experienced a temperature increase of
~0.21 °C decade$^{-1}$ from 1978 to 2002 (Hari et al., 2006).
LB is a 74 m deep, meso-eutrophic, medium-sized lake resting at an elevation of 429 m. It covers a surface area
of 39.3 km$^2$ and holds a volume of 1.18 km$^3$ with hydraulic residence time of 58 days. Complete deep convective
mixing occurs every winter and effectively replenishes the oxygen-depleted deep-water. The Aare River provides
~61% of LB's inflow and experienced a 0.34 °C decade$^{-1}$ increase in temperature from 1978 to 2002 (Hari et al.,





111 2006). Several dams/lakes trap sediment along the upstream Aare course and influence flow in ways that increase

112 sediment settling and water temperature prior to entering LB. The Mühleberg Nuclear Power Plant (MNPP),

113 situated ~19 km upstream from LB (7°16' E, 46°59' N; Fig. 2) represents a point-source of thermal pollution.

114 Planned for decommission in 2019, the plant emits 700 MW of heat into Aare and substantially warms the river

115 water (Råman Vinnå et al., 2017). The ~8 km long Zihlkanal, LB's second largest tributary, supplies ~32% of the

116 lake inflow and connects LB to Lake Neuchâtel (Fig. 2). This tributary is neglected here since it mainly transport

117 lake surface water, which has approximately the same temperature as LB surface water and thus heats equally.

118

## 2.3 River models

### 2.3.1 Temperature

121 Uncertainty concerning river morphology, heat fluxes, shadowing and atmospheric conditions such as wind speed

122 and cloudiness (Caissie, 2006) pose a significant challenge to accurate modelling of future river temperatures.

123 Deterministic models typically require detailed knowledge unavailable for future climate scenarios. Regressions

124 and stochastic models rely heavily on observed natural variability applicable to a given time frame and typically

125 do not include inputs representing additional or interacting physical processes. On their own, these sorts of "black

126 box" models cannot balance trade-offs between constraints available from empirical data and the complexity

127 offered by theoretical frameworks.

128 To overcome these limitations, we used the hybrid model air2stream (Toffolon and Piccolroaz, 2015). The model

129 combines the simplicity of stochastic models with accurate representation of the relevant physical processes

130 affecting temperature. Similar to the neural networks approach, the model calculates river water temperature ($T_w$)

131 through a Monte Carlo-like calibration process, which identifies optimal parameters for weighting physically

132 dependent variables. We used the eight-parameters ($a_1$ to $a_8$) version of the model which incorporates air

133 temperature ($T_a$) and river discharge ($Q$) as a function of time ($t$).

134

$$\frac{dT_w}{dt} = \frac{1}{\delta}\left\{a_1 + a_2 T_a(t) - a_3 T_w(t) + \theta\left[a_5 + a_6\cos\left(2\pi\left(\frac{t}{t_y} - a_7\right)\right) - a_8 T_w(t)\right]\right\};$$

$$\delta = \theta^{a_4}; \theta = Q(t)/\overline{Q};$$

(1)

135 where $t$ is expressed in years and $t_y$ represents one year. Both Aare and Rhône (stations 2085 and 2009,

136 respectively, Fig. 2) provided calibration (1990 – 1999) and validation data (2000 - 2009). Table 1 and Fig. 3

137 show best-fit parameters and model performance statistics. Model sensitivity to variation in $T_w$ was assessed by

138 removing MNPP thermal pollution and repeating the calibration/validation for station 2085 (Table 1).


### 2.3.2 Suspended sediment concentration

141 Water density and intrusion depth of river water into downstream lakes is influenced by SSC. Intensive flow

142 events create high levels of SSCs (Rimmer and Hartmann, 2014), as can exposure/erosion of sediment sources





within the river basin through the so-called hysteresis effect, in which SSC varies for the same level of discharge
(Tananaev, 2012). River discharge regimes have been predicted to change in the future (Birsan et al., 2005),
suggesting that SSCs will also change. To simulate future SSCs, we used the supply-based rating model described
in Doomen et al. (2008), which Fink et al. (2016) adapted to the Rhine River.
The model consists of a base level SSC (g m$^{-3}$) function expanded to express erosion of sediment at high discharge
and sediment accumulation at low discharge. The model is expressed as
$$SSC(t) = m + b_1 Q(t)^{c_1} + d_1 d_2 b_2 \left( Q(t) - Q_{\text{th}} \right)^{c_2} - b_3 \left( 1 - d_2 \right), \tag{2}$$

where $b_x$, $c_x$ and $m$ are adjustable parameters in combination with the threshold discharge ($Q_{\text{th}}$), which determines
whether erosion or deposition occurs within the river. The parameters $d_1$ and $d_2$ control the deposition/erosion
of/from the river sediment storage ($\psi$ (g))
$$d_1 = \begin{cases} 0 : \psi = 0 \\ 1 : \psi > 0 \end{cases} \tag{3}$$

$$d_2 = \begin{cases} 0 : Q \leq Q_{\text{th}} \\ 1 : Q > Q_{\text{th}} \end{cases} . \tag{4}$$

Erosion occurs if $Q$ exceeds $Q_{\text{th}}$ and the river basin contains erodible sediment ($\psi > 0$). Sedimentation occurs if $Q$
is smaller than $Q_{\text{th}}$. The change in $\psi$ over time can be formulated as
$$\frac{\mathrm{d}\psi}{\mathrm{d}t} = \left( b_3 \left( 1 - d_2 \right) - d_1 d_2 b_2 \left( Q(t) - Q_{\text{th}} \right)^{c_2} \right) Q(t) \tag{5}$$

Parameters in equations (2) to (5) were calibrated (2013) and validated (2014) through a evolutionary algorithm
(Fink et al., 2016). Table 2 and Fig. 4 give model performance statistics and best-fit parameter values.

**2.4 Lake model**
We used the one-dimensional model SIMSTRAT (Goudsmit et al., 2002) to assess the impact of climate change
on temperature and deep-water renewal in LB and LG. The model calculates heat fluxes and vertical mixing driven
by wind and the internal wave field. SIMSTRAT has been adapted to and validated for multiple lakes including
Lake Zürich (Peeters et al., 2002), LG (Perroud and Goyette, 2010; Schwefel et al., 2016), Lake Neuchâtel
(Gaudard et al., 2016), Lake Constance (Fink et al., 2014b; Wahl and Peeters, 2014) and LB (Råman Vinnå et al.,
2017). Here we used best-fit parameters already established and validated by Schwefel et al. (2016) for LG and
by Råman Vinnå et al. (2017) for LB (Table 3).
Building upon the model developed by Råman Vinnå et al. (2017), we introduced an extended river intrusion
scheme described in Appendix A1 (including sensitivity analysis). Lake water entrainment into plunging
underflows are modelled as proposed by Akiyama and Heinz (1984) with additional sedimentation of suspended
load (Mulder et al., 1998; Syvitski and Lewis, 1992). The method addresses the transition of a homogenous open



channel flow to a stratified underflow where entrainment and settling of sediment depend on bottom slope angle.
The model scheme consists of (i) the homogenous region where river water makes up the entire water column,
(ii) the plunging region where the plume separates from the lake surface and (iii) the underflow region where the
plume descends downslope while entraining surrounding water until it separates from the bottom and intrudes
into the lake interior (Fink et al., 2016).

**2.5 Data, hydrology and climate forcing**
The models described above used hourly resolved data from 1989 to 2009 as inputs. For calibration/validation of
river temperature, we used flow and temperature data from the Aare monitoring station #2085 (Fig. 2; 7°11' E,
47°3' N) and from the Rhône monitoring station #2009 (Fig. 2; 6°53' E, 46°21' N). The nearest meteorological
stations, Mühleberg (#5530 Fig. 2; 7°17' E, 46°58' N) for Aare and Aigle (#7970 Fig. 2; 6°55' E, 46°20' N) for
Rhône, provided air temperature data. Due to insufficient representation of high turbidity events, we
calibrated/validated the SSC model with turbidity data converted to SSC with suspended load mass samples from
2013 and 2014.
The meteorological data used for SIMSTRAT included air temperature, vapour pressure, wind speed, solar
radiation and cloud cover. These data were collected from the meteorological stations Cressier (#6354 Fig. 2;
7°03' E, 47°03' N) for LB and Pully (#8100 Fig. 2; 6°40' E, 46°31' N) for LG. Råman Vinnå et al. (2017) and
Schwefel et al. (2016) provide additional information on climate data inputs to the one-dimensional model. The
model's river intrusion scheme requires as input the slope angle travelled by the river underflow, which was
obtained from a 25 m resolved digital height model (DHM25). Vertical temperature profiles, sampled at the
deepest location of both lakes in January 1989, were used as initial conditions.
Van Vliet et al. (2013) suggested that river discharge and air temperature should be used while predicting future
river temperatures. We incorporated recent findings of climate-induced changes in air temperature and river
discharge regimes to model both future river temperature and sediment loads. Seasonal mean predictions for air
temperature increase in western Switzerland (Fig. 2), were estimated from CH2011 (2011) under the A1B
emission scenario (balanced use of renewable and fossil fuels) using results from twenty regional climate models.
Flow projections were obtained from published results generated by the PREVAH (PREcipitation-Runoff-
EVApotranspiration HRU Model) hydrological model (Viviroli et al., 2009) using a gridded configuration as
described in Speich et al. (2015) and Kobierska et al. (2011). The model explicitly incorporates changes in glacial
extent, snow melt, catchment runoff, floods and low water flows (FOEN, 2012; Bosshard et al., 2013; Speich et
al., 2015). The PREVAH outcomes for the 1981-2009 period have been validated with data from 65 river gauges
(Speich et al., 2015), including the two gauges upstream of LG (Rhône, #2009 in Fig. 2) and LB (Aare, #2085 in
Fig. 2) used here.

**2.6 Model scenarios**
Six different model scenarios were used to propagate climate change effects through the major tributaries and
their associated downstream lakes. Model scenarios LG1 to LG3 represented LG while LB1 to LB3 represented





LB (Table 4). Each scenario includes three time periods: a reference period (1990-2009), a near-future period
(2030-2049) and a far-future period (2080-2099). The twenty-year intervals allowed us to resolve natural
variations at seasonal and shorter time scales. We initialized the models one year prior to the investigated period
for each time frame (1989, 2029 and 2079) in order to remove effects of initial conditions.
Scenarios LG1 and LB1 excluded river inflow in order to isolate lake response to climate change from potential
tributary influence. Scenarios LG2, LG3, LB2 and LB3 were used to differentiate between the effects of tributary
temperature and SSC, and to provide model sensitivity estimates. The LB3 scenario excluded MNPP thermal
pollution from near-future and far-future time periods but not from the reference period. The LB2 scenario
included thermal pollution in modelling river water temperature. Scenarios LB2, LB3 and LG3 included SSC
while LG2 did not. Low SSC values found in the Aare data resulted in negligible differences between models
including and excluding SSC. Because they served primarily validation and sensitivity analysis purposes, the
Aare/LB model results excluding particles and including/excluding MNPP thermal pollution (LB4 and LB5) are
relegated to Appendix Figure B1 and not discussed further. Scenarios LG3 and LB3 represent expected future
developments.
The unmodified air temperature and modelled river discharge/temperature/SSC were used as inputs for the
reference periods. Near-future and far-future models incorporated predicted changes in air temperature and river
discharge/temperature/SSC with maximum, medium/mean and minimum values serving as envelopes for each
parameter (Figs. 2a and 5). This strategy gave nine simulations (three for scenario LG1 and LB1 which exclude
rivers, i.e. no variation of discharge nor river temperature) for each near-future and far-future time period.
Predicted results included a total of 87 model runs. Upper, mean and lower impact estimates (described and
interpreted below) were derived from the nine basic model runs.

**3 Results**
**3.1 Rivers**
The seasonality of predicted river discharge ($Q$) from FOEN (2012) varies with respect to the reference period
1990-2009 (Figs. 5a and 5b). The PREVAH model show a future decrease of mean summer discharge (1st April
to 30th September) for both the Aare (-3.7 m$^3$ s$^{-1}$ decade$^{-1}$, #2085) and Rhône (-3.8 m$^3$ s$^{-1}$ decade$^{-1}$, #2009). The
decrease in summer will be compensated by an observed increase in winter flow (1st October to 31st March) of the
Aare (+3.3 m$^3$ s$^{-1}$ decade$^{-1}$) and Rhône (+3.7 m$^3$ s$^{-1}$ decade$^{-1}$). These results confirm previous findings presented
in Addor et al. (2014) and Bosshard et al. (2013).
Regional air temperatures from the A1B emission scenario (~+0.32 °C decade$^{-1}$; CH2011, 2011; Fig. 2a) cause a
predicted increase in mean annual water temperature ($T$) for both the Aare (~+0.10 °C decade$^{-1}$) and the Rhône
(~+0.08 °C decade$^{-1}$). Both rivers experience seasonal variations in temperature increase similar to that predicted
for air temperatures (Figs. 2a, 5e and 5f). The effect is strongest in Aare during summer with warming of up to
+2.5 °C in water temperatures for the far-future time period relative to the reference period.
Thermal pollution from MNPP in the Aare during the reference period (blue-green line in Fig. 5e, Råman Vinnå
et al., 2017) causes approximately twice as much heating in winter relative to warming from climate change in





the far-future. In summer, the relationship reverses with minor MNPP warming relative to that induced by climate
change. The net effect of climate warming and MNPP decommission (i.e. removal of MNPP heat from near-future
and far-future time periods) on the Aare is cooling in winter and warming in summer relative to the reference
period (Fig. 5c). Climate change and local anthropogenic thermal input can thus exert similar impacts on river
temperatures, consistent with previous findings by Wright et al. (1999).
Like river temperatures, SSCs depend on river discharge. Our model therefore show SSC increasing in winter and
decreasing in summer due to shifts in discharge regime (Figs. 5g and 5h). The model results for Rhône exhibit a
mean seasonal increase of +14 g m$^{-3}$ decade$^{-1}$ in winter and a decrease of -11 g m$^{-3}$ decade$^{-1}$ in summer. For reasons
explained above (section 2.2), results for the Aare show less variation, with a seasonal increase of +0.3 g m$^{-3}$
decade$^{-1}$ in winter and a decrease of -0.4 g m$^{-3}$ decade$^{-1}$ in summer. Altered temperature and SSC caused increases
and decreases in water density for both rivers in winter and summer, respectively.

**3.2 Lakes**
Warmer air temperatures (Fig. 2a) predicted from climate change resulted in temperature increases in both LG
and LB for all scenarios (Table 5). Models showed the highest warming rates in the epilimnion, intermediate
values throughout the metalimnion and the lowest rates in the hypolimnion (Table 5). We defined the epilimnion,
metalimnion and hypolimnion using the water column stability method described in Råman Vinnå et al. (2017).
The predicted warming of LG varied only slightly among the three-different scenarios (Figs. 6a to 6c). Predicted
warming of LB depend strongly on the scenario used (Figs. 6d to 6f).
Similar to the predicted warming patterns for rivers (section 3.1), both lakes showed seasonally varying warming
patterns. Reduced warming corresponds with periods of high river discharge (Figs. 5a and 5b). This cooling effect
occurs primarily in winter and mid-summer, focussed in depth to the level of river intrusion (Figs. D1b, D1d and
7c to 7f). Model results showed a greater degree of fluctuations of the warming in LB than in LG. This probably
results from the greater influence of the Aare on LB compared to that of the Rhône on LG, as LG has a longer
hydraulic residence time. Scenario LB1, which excludes river intrusion, showed only limited seasonal variation
in warming (Figs. C1c and C1e). According to these results, the closure of MNPP could offset climate-induced
warming of LB by ~25% (~-0.02 °C decade$^{-1}$).
Model results show that enhanced warming of the epilimnion relative to the hypolimnion strengthens stratification
(Figs. 7g and 7h). This enhances the duration of stratification (for both lakes ~+2 days decade$^{-1}$; Table 5) and
slightly elevates the thermocline (in LB ~-0.1 m decade$^{-1}$ and in LG ~-0.05 m decade$^{-1}$; Table 5). We used the
Schmidt (1928) stability ($S$) equation to estimate stratification strength ($S$) (J m$^{-2}$):
$$S = \frac{g}{A_0} \sum_{z=0}^{z_{\max}} (z - z_{\mathrm{m}})(\rho(z) - \rho_{\mathrm{m}}) A(z) \Delta z . \qquad (6)$$

Eq. 6 incorporates gravity ($g$ = 9.81 m s$^{-2}$), depth ($z$), lake surface area ($A_0$), horizontal cross section area ($A(z)$),
lake density ($\rho(z)$), maximum depth ($z_{\max}$), mean lake density ($\rho_{\mathrm{m}}$), lake volume ($V$) and volumetric mean depth
($z_{\mathrm{m}}$) defined as



$$z_{\mathrm{m}} = \frac{1}{V} \sum_{z=0}^{z_{\max}} z A(z) \Delta z \, . \tag{7}$$

The duration of stratification was determined by counting the days when temperature differed by more than 1° C between surface (2 m depth) and deep-water (280 m for LG; 50 m for LB) (Foley et al., 2012). The maximum water column stability expression $N^2 = -(g/\rho)\,\mathrm{d}\rho(z)/\mathrm{d}z$ (s$^{-2}$) was used to determine the thermocline depth.

The river intrusion depth is dependent on water density (temperature and SSC). The Rhône is colder (Figs. 5c and 5d) and carries more suspended sediment (Figs. 5g and 5h) than Aare. Reference period results showed that the Rhône intruded in LG at greater depths relative to depths of the Aare intrusion into LB (Figs. 8 and D1). Given the future change in river temperature and SSC, intrusion patterns will thus change as the densities of both Aare and Rhône increase and decrease during respective winter and summer seasons (section 3.1). This explains model results showing respective deeper and shallower intrusions during winter and summer for both rivers (Fig. D1).

Model results show that warming of the Rhône generally diminishes the amount of river water penetrating beyond 200 m depth in LG (Fig. 8a). Elevated winter SSCs however increase river density. The combined effect of temperature and SSC variation caused an increase in the amount of river water intruding beyond 200 m depth (Fig. 8b). The difference in winter heating for the Aare and LB epilimnion (Figs. 5c, 5e and 6c) generally increased the amount of water penetrating into the hypolimnion (Fig. 8c). Decommission of the MNPP enhances temperature differentials between LB and the Aare, thereby increasing the amount of water reaching the deeper parts of LB (Fig. 8d). In summary, the change in river discharge regime for the Aare and Rhône results in respective increase and decrease in winter and summer water density, resulting in a summer to winter shift of the amount of river water penetrating deeper than the metalimnion for both lakes.

**4 Discussion**

**4.1 Rivers**

Increases in air temperature expected from climate change modify tributary runoff. Less water is predicted to be bound in snow and ice at high elevation during winter and spring/summer floods occur earlier (CH2011, 2011; FOEN, 2012). The changed river discharge regime, appearing as increased flow in winter and decreased flow in summer (Figs. 5a and 5b), amplifies the increase and decrease in river temperature during respective summer and winter periods (Figs. 5e and 5f). Amplification results from (i) a smaller flow volume requiring less energy to heat and (ii) lower flow velocities which extend heat exposure. The PREVAH model predict that the future discharge of Aare in summer will be ~20% less than summer discharge predicted for the Rhône. Results therefore suggest that the Aare summer conditions will be more impacted by climate change than Rhône summer conditions (Figs. 5e and 5f).

Model results concerning discharge-dependent responses to climate-induced warming were consistent with previous findings reported by Isaak et al. (2012) and van Vliet et al. (2013). The river temperature increases predicted by this study (0.10° C decade$^{-1}$ for the Aare and 0.07° C decade$^{-1}$ for the Rhône) were much smaller than past observed warming rates (0.34° C decade$^{-1}$ for the Aare and 0.21° C decade$^{-1}$ for the Rhône; Hari et al., 2006).




These differences may reflect contrasting reference periods with past observations conducted from 1971 to 2001
and modelled observations addressing 1990 to 2099. Past observations also incorporate effects of solar brightening
during the 1980s (Fink et al., 2014a; Sanchez-Lorenzo and Wild, 2012; Wild et al., 2007).
Climate change effects aside, MNPP decommissioning in 2019 is predicted to decrease the temperature in the
Aare by up to 4.5°C at station #2085 (Råman Vinnå et al., 2017). The cooling effect of this plant closure primarily
affects winter conditions when climate change induced warming is weaker and river flow is lower (Fig. 5e). The
heating of Aare and LB by MNPP heat emissions equates to approximately one decade of climate-induced
warming of lake surface waters (O'Reilly et al., 2015; Råman Vinnå et al., 2017). This result highlights the role
of point source thermal contributions in local climate impact assessments. The effect is highly seasonally
dependent but persists into far-future time periods (Figs. 5c and 5e).
The amount of suspended sediment carried by rivers depends on both discharge and the amount of erodible
sediment in the watershed (Fink et al., 2016). We used a supply-based sediment rating model subjected to a
changing discharge regime to examine changes in suspended sediment from summer to winter for both Aare and
Rhône (Figs. 5g and 5h). Consistent with previous findings reported by Pralong et al. (2015), we predict an
increase in SSC during winter and decrease of SCC in summer.
Figure 4 and Table 2 show that the SSC model gives robust results for Rhône (coefficient of determination $R^2$ =
0.68 from 2013 to 2014) but not for Aare ($R^2$ = 0.06 from 2013 to 2014). The Aare includes several sediment-
trapping reservoirs/lakes upstream of station #2085. Peaks in SSC at station #2085 thus do not reflect watershed-
scale discharge events (Fig. 4) but rather local precipitation and discharge events in the headwaters of the Saane
River, a tributary to Aare (Fig. 2). This tributary hosts few sediment traps and contributes ~34% of the downstream
flow at station #2085. Given the limited impact of SSC on Aare water density, models show only negligible impact
on river intrusion depth and corresponding intruding volumes (Figs. 8c, B1c, B1e and D1c). The lower reaches of
the Rhône are not dammed, thus adhering more directly to model assumptions and giving clearer results (Fig. 4).
High SSC events are usually associated with extreme floods (Fink et al., 2016), which are predicted to vary in
alpine lake catchments with on-going climate change (Glur et al., 2013). The lack of constraints on extreme
precipitation events introduces uncertainty into future flood frequency and magnitude predictions (CH2011,
2011). Shifts in river discharge regimes also depend on the amount of water bound in snow and ice as well as on
the timing of spring/summer melt. Future climate scenarios predict that ~30% of the glacier mass will remain in
the Aare and Rhône catchments by the end of the 21st century (FOEN, 2012). Glacial meltwater is thus expected
to continue to supply the Aare and Rhône throughout the time frames considered in this study. We thus assumed
natural flood frequencies for our reference period and then adjusted them for overall river discharge regime shifts
predicted by FOEN (2012).

**4.2 Lakes**
All model scenarios showed that increased air temperature leads to warming of both lakes, especially of the
epilimnion (Table 5, Fig. 6). Piccolroaz et al. (2015) showed that an increase of lake stability and earlier onset of
stratification causes warming of surface waters due to the smaller volume undergoing warming and diminished



heat transfer to the hypolimnion. The lake model used here showed an increase of stratification strength and a
lengthening of the stratified period in both lakes (Table 5, Figs. 7g and 7h). Our results thus support consistently
previous findings for LG reported by Foley et al. (2012), O'Reilly et al. (2015) and Schwefel et al. (2016).
Seasonal variation in warming of both epilimnion and hypolimnion (Figs. 7a to 7f) exceeded that expected from
future changes in air temperature (Fig. 2a). The model showed a decrease in warming during winter and mid-
summer, which corresponds in time to periods of high river discharge from the main tributaries (Figs. 5a and 5b).
This cooling effect was more effective for LB than for LG (Fig. 7) and appeared in all scenarios except for LB1
and LG1 (Fig. C1), both of which exclude coupled river effects. The extended seasonal variation in climate
warming is thus driven by river discharge volume and temperature trends (Figs. 5 and 7). This response applies
to aquatic systems in which a difference exist in temperature and heating regimes between rivers and lakes, but
does not appear to affect water bodies with uniform temperature/heating regimes. Our results thus supports the
hypothesis put forward by Zhang et al. (2014), stating that climate warming of lakes might be reduced and even
reversed by addition of external water.
To investigate this effect, we varied the hydraulic residence time of LB and LG, while holding all other factors
constant (Fig. 9). We implemented a stepwise reduction in LG size (to 1/80 of its original volume), simultaneously
reducing hypsographic area but keeping maximum depth unchanged. Similar adjustments was made to LB to
obtain corresponding hydraulic residence times. This stepwise approach required 972 additional model runs.
These iterations showed that river water had to be cooler than lake water in order to generate the climate warming
dampening effect (Figs. 9a and 9d). Penetration into deep lake reaches by large river volumes strengthens the
effect in the hypolimnion (Fig. 9b). The climate dampening effect is suppressed when river and epilimnion
temperature are similar. MNPP thermal input creates such conditions in the Aare and therefore largely counteract
the river cooling effect of Aare on LB (Fig. 9c). For shorter residence times (< ~1000 days), rivers can exert
influence if a significant temperature difference exists between river and lake waters. For longer residence times
(> ~1000 days), tributaries cannot significantly offset climate effects in downstream water bodies.
Climate-induced warming of lakes (Schwefel et al., 2016), along with changing frequency or intensity of deep
penetrating flood events (Fink et al., 2016) may curtail oxygen supply to deep lakes. Recent flood analysis has
also indicated that input of river-borne organic matter increases respiration, causing a paradoxical net oxygen
deficit within the intruding layer (Bouffard and Perga, 2016). These findings highlight the necessity of estimating
the amount of oxygen saturated river water entering the hypolimnion. We therefore evaluated models
implementing climate-caused shifts in river discharge regimes for reaction of lakes and rivers to climate-induced
warming, shifts in seasonal SSC in rivers and local thermal pollution effects. Models showed respective winter
increase and summer decrease in river water density relative to lake stability. This creates summer to winter
seasonal shifts in deep intrusion dynamics for both lakes (Fig. D1). Models showed a net annual effect of increased
volumes of river water penetrating into deeper parts of both lakes (Fig. 8). An increase in winter SCC represented
the primary driver for increases in the amount of water annually penetrating past 200 m depth in LG (Figs. 8a, 8b
and D1a, D1b). In LB, colder winter river temperatures (relative to lake temperatures) due to removal of MNPP
heat contributed to greater penetration (Figs. 8c, 8d and D1c, D1d).





Fink et al. (2016) also found evidence that climate change will cause diminished deep river intrusion events in
summer and enhanced intrusion in winter. They predicted an annual decrease in the amount of river water reaching
the deepest parts of Lake Constance. The tributaries considered here differ from the Rhine River investigated by
Fink et al. (2016) primarily in terms of their temperature. The Rhône catchment for example rests at a mean
elevation of 2127 m and includes greater glacial coverage (11%) whereas the Rhine catchment rests at mean
elevation 1771 m and only has 1% glacial coverage. The closure of the MNPP and associated temperature decrease
contribute to increase of the volume/frequency of deep intrusion (Fig. 5). While Fink et al. (2016) focused
primarily on flood frequencies, our models emphasized river discharge regimes and interacting river and lake
temperature regimes. The annual increase in river penetration to depth predicted by our models suggests future
increase of deep-water oxygen supply in similar tributary-lake systems. This prediction applies primarily to
meromictic lakes such as LG. Analogous effects in holomictic lakes such as LB, which mix completely each
winter, are less significant. Similar to findings of Fink et al. (2016), our models indicate that deep-water oxygen
conditions will worsen during strongly stratified conditions due to seasonal shifts in deep river intrusions from
summer to winter.

### 406   4.3 Model reliability

Predictions concerning the effect of climate change on rivers and lakes depend on i) the choice of emission
scenario, ii) the accuracies of models linking climate to hydrology and climate to heat fluxes  and iii) natural
variability of the system being investigated (Raymond Pralong et al., 2015). This section describes uncertainties
and limitations of our approach.
Results of long-term forecasts (beyond 2050) depend strongly on representations of global greenhouse gas (GHG)
emission scenarios (FOEN, 2012). Given uncertainties in future global climate policy, we chose a median
scenario, which falls between best (ex. RCP3PD) and the worst case scenarios (e.g. A2) in terms of GHG
emissions. A1B assumes a peak population at mid-century, balanced use of renewables and fossil fuels and rapid
introduction of new technologies.
Estimates of future air temperatures and river discharge were obtained from a combination of regional climate
models (RCMs; CH2011, 2011; FOEN, 2012). Uncertainties associated with individual RCMs were offset by
combined forecasts from multiple-model chains. Numerous studies have performed detailed evaluations of
uncertainty in air temperature and river discharge under established emission scenarios (RCP3PD, A1B, A2) and
accounting for global-regional climate model interactions (Addor et al., 2014; Bosshard et al., 2011, 2013;
CH2011, 2011).
The degree of accuracy with which model input parameters represent future conditions determines the accuracy
of model predictions. We therefore ran the river temperature model with varying parameters to evaluate model
sensitivity (Table 1) for different yet similar datasets. The air2stream parameter $a_1$ showed the greatest degree of
sensitivity, varying within three orders of magnitude. The $a_1$ parameter, however, does not respond to variation in
river discharge or air temperature (Equation 1), which limits its sensitivity to climatic input data. The other
parameters ($a_2$ to $a_8$) varied only within one order of magnitude (Table 1). The SSC model gives better results for
the Rhône (coefficient of determination $R^2 = 0.68$ from 2013 to 2014) than for the Aare ($R^2 = 0.06$ from 2013 to



2014). Dam and reservoir infrastructure upstream of station #2085 along the Aare dampen sediment transport
events and decouple them from regional discharge events (see above; Fig. 4). Given the relatively minor effect of
SSC on Aare water density, variation in the input parameter does not influence river intrusion depth estimates
(Figs. B1e to B1f and D1c to D1d). Previous studies have evaluated the accuracy of SIMSTRAT lake model
predictions based on climatic data (Schwefel et al., 2016) and anthropogenic thermal emissions (Fink et al., 2014b;
Råman Vinnå et al., 2017). As with other vertical, one-dimensional models, SIMSTRAT cannot account for lateral
heterogeneities in lakes. This inherent weakness in model design however does not significantly diminish the
accuracy of model predictions concerning LB and LG (Råman Vinnå et al., 2017; Schwefel et al., 2016).
This study assumes that glacial melt-water feeding both the Aare and Rhône in summer will not disappear within
the time frames considered here. Loss of glacial sources would drastically modify the assumed discharge regime,
especially in summer, which would affect accuracy of temperature, SSC and intrusion depth estimates presented
here. However, as stated in section 4.1, FOEN (2012) predicts that the Aare and Rhône catchments will retain
30% of their glacial masses by the year 2100. These predictions support assumptions concerning the Aare and
Rhône discharge regimes used here. Point sources/sinks of anthropogenic heat can affect inland water bodies
response to climate change, as shown by the MNPP effects described here. Other changes in catchment
management, such as hydropower dam construction would also likely alter river discharge regimes and by
extension, temperatures, SSCs and deep-water renewal (Fink et al., 2016). The accuracy of future climate change
predictions depends on accurate accounting of regional anthropogenic factors affecting physical processes in the
system under investigation.

**5 Conclusion**
Aquatic processes in lakes are the result of regional climate and events in upstream tributaries. This study
investigated the impact of climate change on inland waters by propagating climatic inputs through integrated
fluvial-lacustrine systems. We entered predicted future climatic data into models for two connected river and lake
systems in order to evaluate downstream thermal responses and how river discharge regime shifts might affect
deep-water renewal in the lakes. Climate data propagated through discharge-dependent river temperature and
suspended sediment concentration models coupled to a one-dimensional lake model yielded theoretical hydrologic
predictions for two peri-alpine lakes, Lake Biel and Lake Geneva.
The models showed that climate warming of rivers is enhanced in summer and diminished in winter due to future
river discharge regimes with decreased flow in summer and increased flow in winter. This climate-caused
alteration of the flow regime likewise increases the river-borne suspended sediment load in winter and decreases
it in summer.
Both lakes showed large seasonal temperature increases that could not be solely explained by climate-related
(predicted) increases in air temperature. Instead, the lakes experienced a cooling effect associated with upstream
tributaries, whose response to increasing future air temperatures differed from that of the lakes. The smaller Lake
Biel showed increased response to this repressive effect on climate warming than that of the larger Lake Geneva.
Predicted changes in Lake Biel strongly depend on removal of upstream anthropogenic thermal emission into the
Aare River. Local anthropogenic point sources of heat can thus rival climate change in terms of their influence on





lakes. This dampening of climate warming depends on the lakes hydraulic residence times and requires adequate
river/lake temperature differences. Our models indicate that tributaries can exert system-wide influence on lakes
with hydraulic residence times < ~1000 days. Lake systems with longer residence times are resistant to tributary
effects but may respond to local effects.
The combination of changes in river SSC and differential lake/river temperature/warming result in a seasonal shift
of deep-water penetration (by rivers) into lakes. The volume of river water penetrating to deeper parts of lakes
specifically decreases in summer and increases in winter. The net effect of these predicted shifts enhances annual
deep-water renewal. Higher rates of deep-water renewal can in turn enhance reoxygenation of the deepest reaches
of lakes, which may otherwise experience lower oxygen concentrations under climate change.

**Data availability**
See acknowledgments

**Appendices**
**Appendix A1 River Intrusion Model**
Figure A1 summarizes the river intrusion model. The depth where the river plume separates from the surface, the
so-called plunge depth ($h_p$), depends on the slope angle ($\sigma$), gravity ($g$), coefficients ($S_1 = 0.25$; $S_2 = 0.75$; Ellison
and Turner, 1959), bed friction ($f_t = 0.02$; Akiyama and Heinz, 1984), initial flow per unit width ($q_0$) = $V_r$ / $W_r$
dependent on river discharge ($V_r$), river width ($W_r = 100$ m for the Aare and $W_r = 120$ m for the Rhône) and the
relative density difference ($\Delta\rho = (\rho_r - \rho_l(z_1)) / \rho_l(z_1)$) between the homogenous river ($\rho_r$) and lake ($\rho_l$) with $z_1 =$
surface.
$$h_\text{p} = e_1 \left( \frac{f_\text{t}}{\sigma(z) S_2} \frac{q_0^2}{g\Delta\rho} \right)^{1/3} + e_2 \left( \frac{q_0^2}{S_1 g\Delta\rho} \right)^{1/3} \tag{A1}$$

The level of initial plume entrainment is treated differently on a gentle versus a steep slope. This is controlled by
the coefficients $e_1$ and $e_2$.
$$e_1 = \begin{cases} 1 : \sigma(z_1) < \sigma_c \\ 0 : \sigma(z_1) \geq \sigma_c \end{cases} \tag{A2}$$

$$e_2 = \begin{cases} 0 : \sigma(z_1) < \sigma_c \\ 1 : \sigma(z_1) \geq \sigma_c \end{cases} \tag{A3}$$

where the critical slope angle ($\sigma_c$) = $f_t S_1 / S_2$ distinguishes between gentle and steep slope designations. The initial
height of the underflow ($h_d$) can then be written as


$$h_{d}(z_1) = h_{p}(1+\gamma)$$ (A4)
where $\gamma$ is the entrance mixing coefficient equal to ~0 for gentle slopes and increasing to larger values for steeper
slopes. Here we find that a value of 0.1 provides best result. The initial underflow temperature ($T_u$), velocity ($U_u$),
particle content ($P_u$) and volume ($V_u$) is consequently expressed as a function of ambient lake water temperature
($T_l$), river temperature ($T_r$) and river particle content ($P_r$) in the homogenous region.
$$T_{u}(z_1) = T_{l}(z_1)\frac{(h_{d}(z_1)-h_{p})}{h_{d}(z_1)} + T_{r}\frac{h_{p}}{h_{d}(z_1)}$$ (A5)
$$U_{u}(z_1) = (1+\gamma)\frac{q_0}{h_{d}(z_1)}$$ (A6)
$$P_{u}(z_1) = P_{r}\frac{h_{p}}{h_{d}(z_1)}$$ (A7)
$$V_{u}(z_1) = V_{r}\frac{h_{p}}{h_{d}(z_1)}$$ (A8)
Once the plume has passed through the plunge region into the underflow region, we express $h_d$, $U_u$, $T_u$ and $V_u$ as
$$h_{d}(z+1) = h_{d}(z) + E(z)\mathrm{d}x$$ (A9)
$$U_{u}(z+1) = U_{u}(z)\frac{h_{d}(z)}{h_{d}(z+1)}$$ (A10)
$$T_{u}(z+1) = T_{l}(z)\frac{(h_{d}(z+1)-h_{d}(z))}{h_{d}(z+1)} + T_{u}(z)\frac{h_{d}(z)}{h_{d}(z+1)}$$ (A11)
$$V_{u}(z+1) = V_{u}(z)\frac{h_{d}(z)}{h_{d}(z+1)}$$ (A12)
where $\mathrm{d}x$ is the horizontal distance between $z$ and $z + 1$ and the entrainment factor ($E$) is expressed as a function
of the entrainment constant ($\beta = 0.0015$; Ashida and Egashira, 1975) and the Richardson number ($R_i$).

$$E(z) = \frac{\beta}{R_{i}(z)}$$ (A13)
$$R_{i}(z) = \frac{f_t}{\sigma(z)S_2}$$ (A14)



For $P_u$, we include a sedimentation term as proposed by Syvitski and Lewis (1992), which depends on the removal
rate ($r$) and $dt = dx / U_u(z)$.

$$P_u(z+1) = P_u(z)\frac{h_d(z)}{h_d(z+1)} - re_3 h_d(z) P_u(z) e^{-r dt} \qquad (A15)$$

Sedimentation only occurs if the plume velocity drops below a critical settling velocity ($U_c$) subject to the
parameter $e_3$:

$$e_3 = \begin{cases} 1 : U_u(z) < U_c \\ 0 : U_u(z) \geq U_c \end{cases} \qquad (A16)$$

We set $U_c$ equal to 0.46 m s$^{-1}$ and $r$ equal to 4.7 day$^{-1}$ to represent medium-sized silt following Mulder et al. (1998).
The plume travels downslope as long as the underflow plume density ($\rho_u$) exceeds $\rho_l(z)$. Once $\rho_u \leq \rho_l(z)$, the plume
raise from the slope and intrudes into the lake proper. The terms $T_u$ and $V_u$ were thus added to the lake model at
this depth. Calculations excluded expressions for the settling of accumulated particles following plume intrusion,
assuming that these exert only minor impacts on lake temperature and density.
The sensitivity of the river intrusion depth to entrainment of ambient water into the plume was tested by
propagating a range of $\beta$ (Eq. A13) values from 1 to $1*10^{-6}$ through model spaces composed of temperature,
discharge and suspended sediment concentration data from Aare (station #2085). Figures A2 to A4 compare
modelled intrusion depths to empirical estimates based on vertical temperature and light transmission data at the
centre of LB (7°11' E, 47°6' N) collected shortly after major river intrusion events. Comparison of the modelled
intrusion depth with light transmission depth (whose minimum value represents a proxy for actual river intrusion
depth) suggests that $\beta = 0.0015$ offers an adequate representation of intrusion depth. Larger $\beta$ values generate
intrusion depths shallower than the empirical reference points whereas smaller $\beta$ values exerted only minor impact
on deepening the intrusion depth.

**Author contributions**
Love Råman Vinnå designed this study and preformed the modelling work; Alfred Wüest provided the founding;
Damien Bouffard contributed to the analysis of the result; Gabriel Fink adapted, calibrated and validated the SSC
model; Massimiliano Zappa provided river dishrag predictions from the PREVAH model; all authors have
contributed to the editing of this manuscript.

**Competing interests**
Do not use Marco Toffolon or Frank Peeters as reviewers, since they are co-authors with Alfred Wüest in recent
publications.





**Special issue statement**
Do not include in special issue.

**Acknowledgments**
This study is part of the "*Hydrodynamic Modelling of Lake Biel for Optimizing the Ipsach Drinking Water Intake*"
project funded by Energy Service Biel (ESB). We are especially thankful to Andreas Hirt, Roland Kaeser and
Markus Wyss for constructive collaboration. We thank the Office of Water Protection and Waste Management of
the Canton of Bern (GBL/AWA) for providing their vertical CTD profiles (data available at
http://www.bve.be.ch/bve/de/index/wasser/wasser/messdaten.html), the Swiss Federal Office of Meteorology and
Climatology (MeteoSwiss) for providing meteorological data (data available at
http://www.meteoswiss.admin.ch/home/services-and-publications/beratung-und-service/data-portal-for-
teaching-and-research.html), the Hydrology Department of the Swiss Federal Office for the Environment (FOEN)
for providing tributary data, the Climate Change and Hydrology in Switzerland (CCHydro) project for providing
future river discharge predictions (available at http://www.bafu.admin.ch/umwelt/index.html?lang=en) and the
Swiss Federal Office of Topography (SwissTopo) for providing DHM25 model bathymetry data (available at
https://shop.swisstopo.admin.ch/en/products/height_models/dhm25). We thank Bettina Schaefli at the University
of Lausanne, Marco Toffolon and Elisa Calamita at the University of Trento, Robert Schwefel at École
Polytechnique Fédérale de Lausanne, Adrien Gaudard at the Swiss Federal Institute of Aquatic Science and
Technology and Stan Thorez at the University of Eindhoven for valuable insights and help with model setup. We
furthermore thank Kei Ito (http://jfly.iam.u-tokyo.ac.jp/html/color_blind/) for valuable feedback on how to adapt
our figures for colour-blind readership.





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





**Tables**
**Table 1.** Air2stream river temperature model best-fit parameters and model
performance statistics reported as coefficients of determination ($R^2$) and
root mean square deviation (RMSD). Input parameters used in this study
are shown in bold-faced type. The model was calibrated, validated and
subjected to sensitivity tests using data from station #2085 (Aare River)
representing past observed conditions and future predicted conditions
assuming MNPP removal (No MNPP).

| Parameter (Unit) | Aare (#2085) | | Rhône (#2009) |
|---|---|---|---|
| | Measurements | No MNPP | Measurements |
| $a_1$ (°C s$^{-1}$) | **2.0316** | 0.6434 | **1.4927** |
| $a_2$ (s$^{-1}$) | **0.2299** | 0.3855 | **0.2774** |
| $a_3$ (s$^{-1}$) | **0.2267** | 0.3177 | **0.4133** |
| $a_4$ (-) | **0.0157** | 0.5622 | **0.6399** |
| $a_5$ (°C s$^{-1}$) | **6.7022** | 16.2387 | **6.4792** |
| $a_6$ (°C s$^{-1}$) | **4.4950** | 9.9855 | **2.3224** |
| $a_7$ (-) | **0.6066** | 0.6066 | **0.5244** |
| $a_8$ (s$^{-1}$) | **0.7156** | 1.5930 | **1.0760** |
| | $R^2$ (-) | | |
| Calibration [a] | 0.97 | 0.96 | 0.94 |
| Validation [b] | 0.95 | 0.96 | 0.94 |
| | RMSD (°C) | | |
| Calibration [a] | 0.83 | 0.95 | 0.52 |
| Validation [b] | 1.02 | 1.06 | 0.59 |

[a] 1990 - 1999
[b] 2000 - 2009



**Table 2.** River suspended sediment concentration (SSC) model best-fit parameters and model performance statistics reported as coefficients of determination ($R^2$) and root mean square deviation (RMSD).

| Parameter (Unit) | Aare (#2085) | Rhône (#2009) |
|---|---|---|
| $m$ (g m$^{-3}$) | 8.8000 | 1.0000 |
| $b_1$ (g s m$^{-6}$) | 0.2650 | 0.0006 |
| $c_1$ (-) | 0.6500 | 2.3200 |
| $b_2$ (g s m$^{-6}$) | 0.0011 | 0.0010 |
| $c_2$ (-) | 2.3000 | 12.0000 |
| $b_3$ (g m$^{-3}$) | 8.8000 | 2.0000 |
| $Q_{th}$ (m$^3$ s$^{-1}$) | 401 | 232 |
| $R^2$ (-) | | |
| Calibration [a] | 0.20 | 0.74 |
| Validation [b] | 0.03 | 0.58 |
| RMSD (g m$^{-3}$) | | |
| Calibration [a] | 82 | 206 |
| Validation [b] | 217 | 222 |

[a] 2013

[b] 2014





**Table 3**. One-dimensional lake model
SIMSTRAT best-fit parameters and model
performance statistics reported as vertical
volume-weighted averaged root mean square
deviation (RMSD-V).

| Parameter (Unit) | Lake Biel | Lake Geneva |
|---|---|---|
| $p_1$ (-) | 1.30 | 1.09 |
| $p_2$ (-) | 1.20 | 0.90 |
| $K$ (-) | 0.70 | 1.40 |
| $q$ (-) | 1.30 | 1.25 |
| $C_{Deff}$ (-) | 0.0050 | 0.0020 |
| $C_{10}$ (-) | 0.0016 | 0.0017 |
| $a_s$ (-) | 0.0060 | 0.035 |
| $a_w$ (-) | 0.0040 | 0.009 |
| RMSD-V (°C) | | |
| Calibration | 0.73 [a] | 0.66 [c] |
| Validation | 0.68 [b] | |

[a] 1995 - 2004
[b] 2005 - 2015
[c] 1981 – 2012

**Table 4.** Model scenarios of climate change effects for near-future and far-future time
periods, including (Inc.) and excluding (Exc.) the effects of rivers and suspended
sediment. Thermal input from MNPP also included/excluded. Most likely scenarios
shown in bold.

| Lake | Exc. Rivers | Inc. Rivers | |
|---|---|---|---|
| | | Exc. Suspended Sediment | Inc. Suspended Sediment |
| Geneva | LG1 | LG2 | **LG3** |
| | | Inc. MNPP | Exc. MNPP |
| Biel | LB1 | LB2 | **LB3** |




**Table 5**. Change in temperature, length of the stratified period and depth of the thermocline
(negative values correspond to a shallower thermocline) for each scenario listed in Table 4.
Estimates given as mean of the daily difference between the reference period and the far-future time
period. Temperature anomalies are volume-weighted and vertically averaged. Most likely scenarios
shown in bold.

| Scenario | Temperature ($°C$ decade$^{-1}$) | | | Stratification (days decade$^{-1}$) | Thermocline (m decade$^{-1}$) |
|---|---|---|---|---|---|
| Lake Biel | Epilimnion | Metalimnion | Hypolimnion | | |
| LB1 | 0.19 | 0.16 | 0.13 | 1.5 | -0.02 |
| LB2 | 0.15 | 0.13 | 0.06 | 2.0 | -0.07 |
| **LB3** | **0.13** | **0.11** | **0.05** | **2.2** | **-0.13** |
| Lake Geneva | | | | | |
| LG1 | 0.17 | 0.13 | 0.07 | 2.9 | -0.07 |
| LG2 | 0.17 | 0.12 | 0.07 | 2.8 | -0.06 |
| **LG3** | **0.18** | **0.16** | **0.08** | **2.2** | **-0.04** |







**Figures**

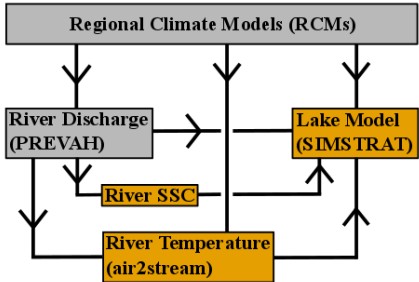

**Figure 1.** Schematic illustration of this study's one-way model chain. Orange models represent modeling performed by this study while grey models represent simulated data inputs obtained from external sources.

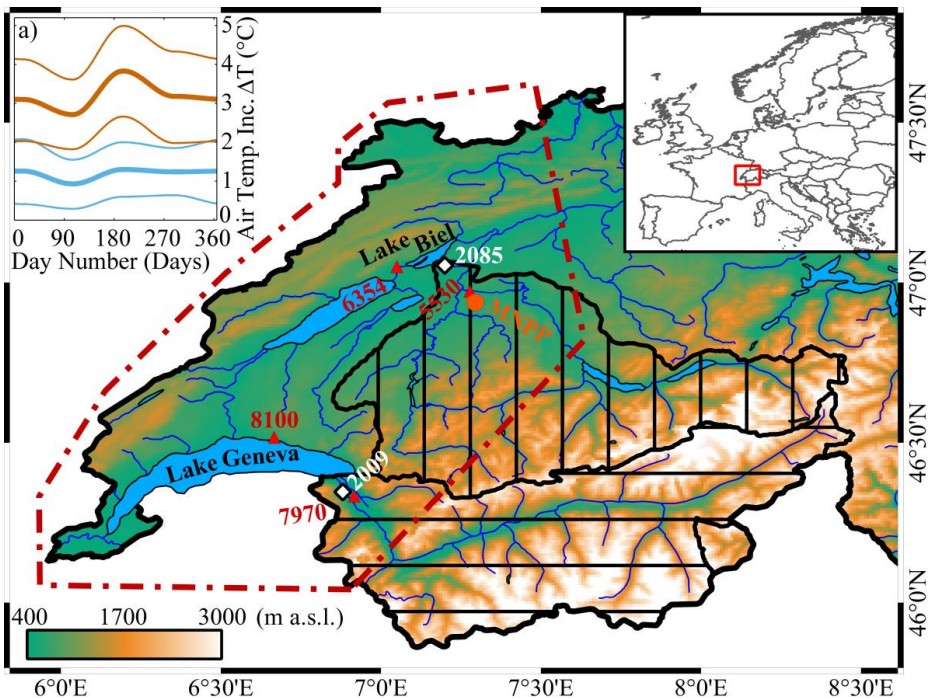

**Figure 2.** Study area and regional air temperature increases predicted under climate change scenarios addressed here. Elevation above sea level (green to white color ramp), locations and number of river stations (white diamonds), drainage area (Aare: vertical lines; Rhône: horizontal lines) and location of Mühleberg Nuclear Power Plant (MNPP, orange circle). Area covered by regional climate models (dark red dashed-dotted line) with a)




predicted air temperature increase ΔT in the near-future (blue, 2030-2049) and far-future (orange, 2080-2099) for
medium (thick lines) and upper/lower estimates (thin lines) under the A1B emission scenario (CH2011, 2011).

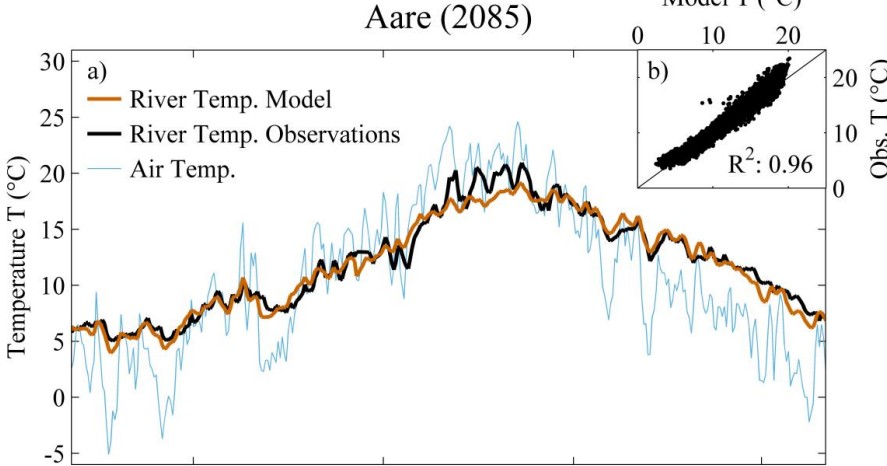

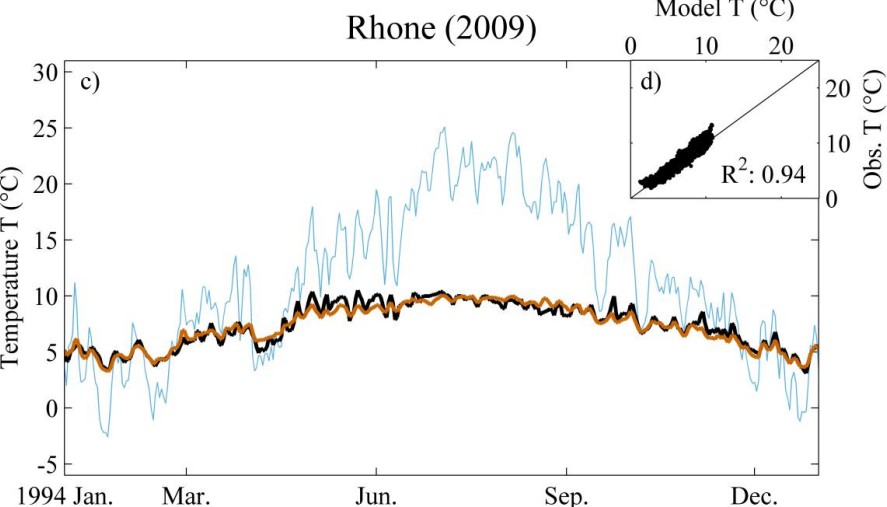


**Figure 3.** Air2stream modeled (orange) and measured (black) temperature (T) compared to air temperature (blue)
for a) Aare River and c) Rhône River in 1994. The insets b) and d) show modeled versus observed temperature
from 1990 to 2009 with coefficient of determination ($R^2$).






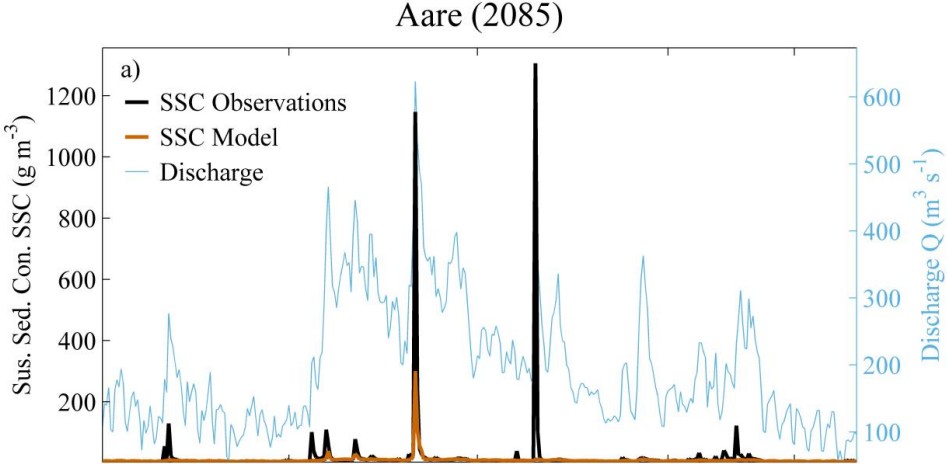

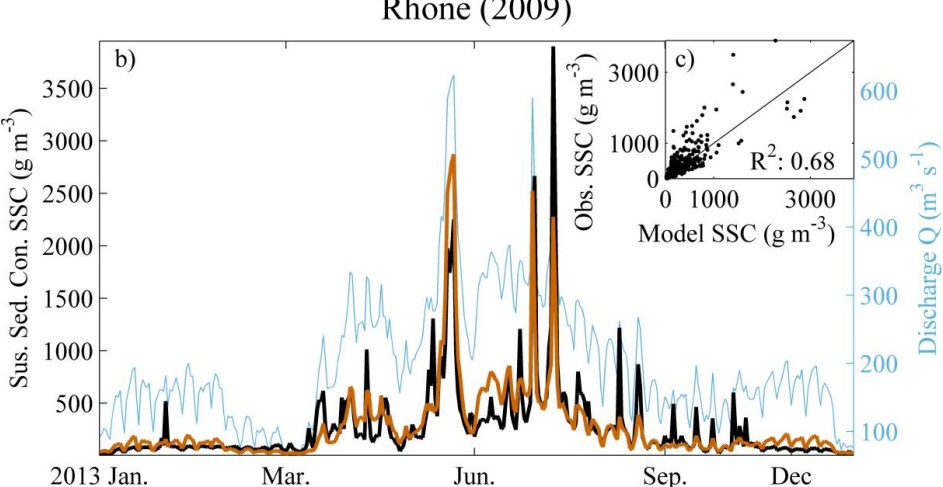


**Figure 4.** Modeled (orange) and measured (black) suspended sediment concentration (SSC) compared to river
discharge Q (blue) for a) Aare River and b) Rhône River in 2013. The insets c) shows modeled versus observed
SSC for 2013 and 2014 with coefficient of determination ($R^2$).



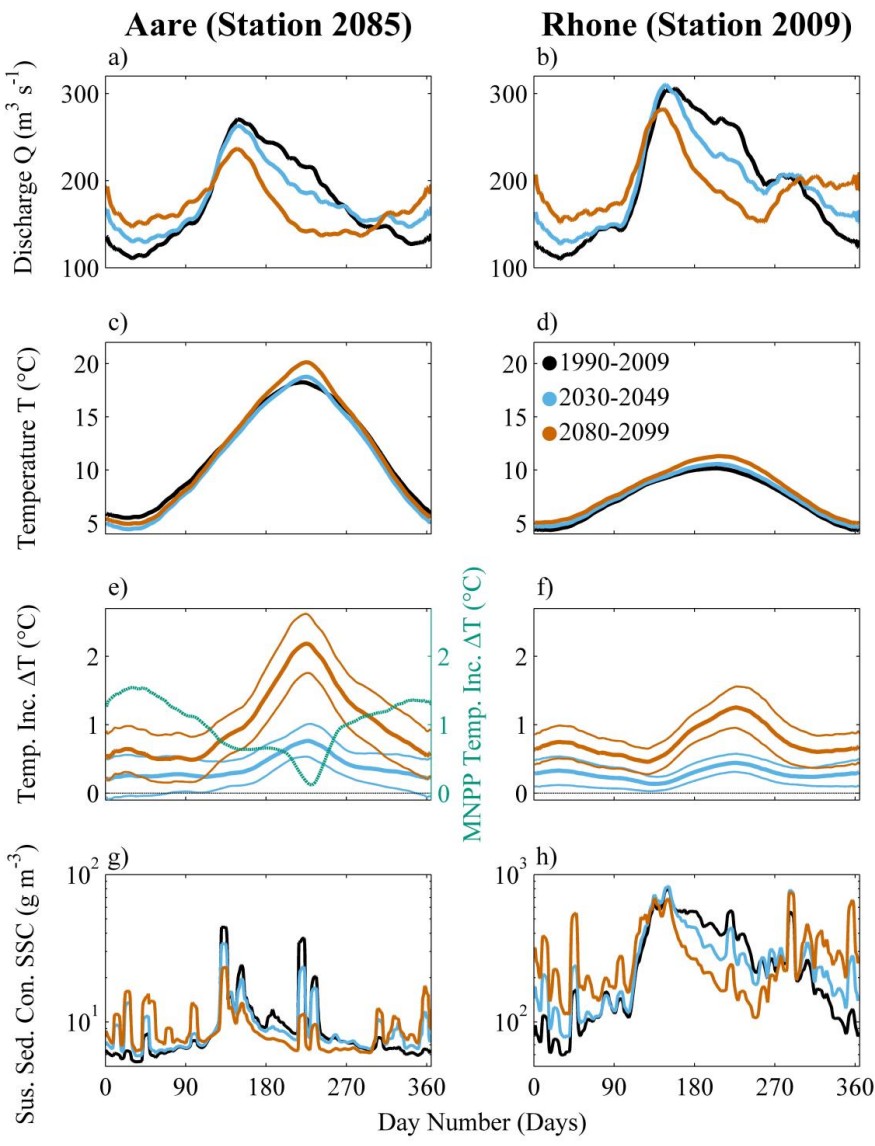

**Figure 5.** Modeled climate impact from scenarios LB3 (Aare River, left column) and LG3 (Rhône River, right column) displayed as daily average for reference (black, 1990-2009), near-future (blue, 2030-2049) and far-future (orange, 2080-2099) time periods. Discharge Q (a and b), net water temperature T (c and d) with anthropogenic heat from Mühleberg Nuclear Power Plant (MNPP) removed from near-future and far-future time periods, temperature increase ΔT (e and f) due to climate change (orange/blue) and MNPP (blue-green) as well as modeled SSC (g and h). Maximum and minimum modeled values are marked by fine lines (e and f) and or are omitted (c, d, g, and h) for clarity.



784

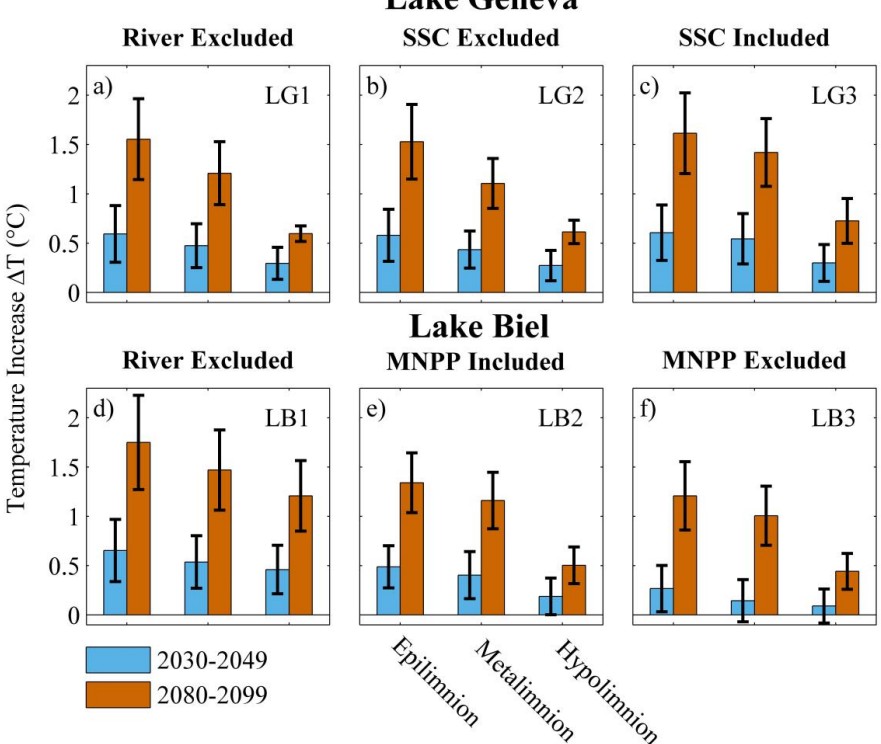

785

**Figure 6.** Temperature increase ΔT for near-future (blue) and far-future (orange) time periods relative to reference period temperatures, displayed as mean (bars) and standard deviation (black lines). Epilimnion (left pair of bars), metalimnion (middle pair) and hypolimnion (right pair) in LG (a to c) and LB (d to f). Graphs represent river intrusion excluded (a and d), river-borne SSC included (c, e and f) and excluded (b), Mühleberg Nuclear Power Plant (MNPP) heat release included in (e) and excluded (f) from near-future and far-future time periods but retained for the reference period.

792



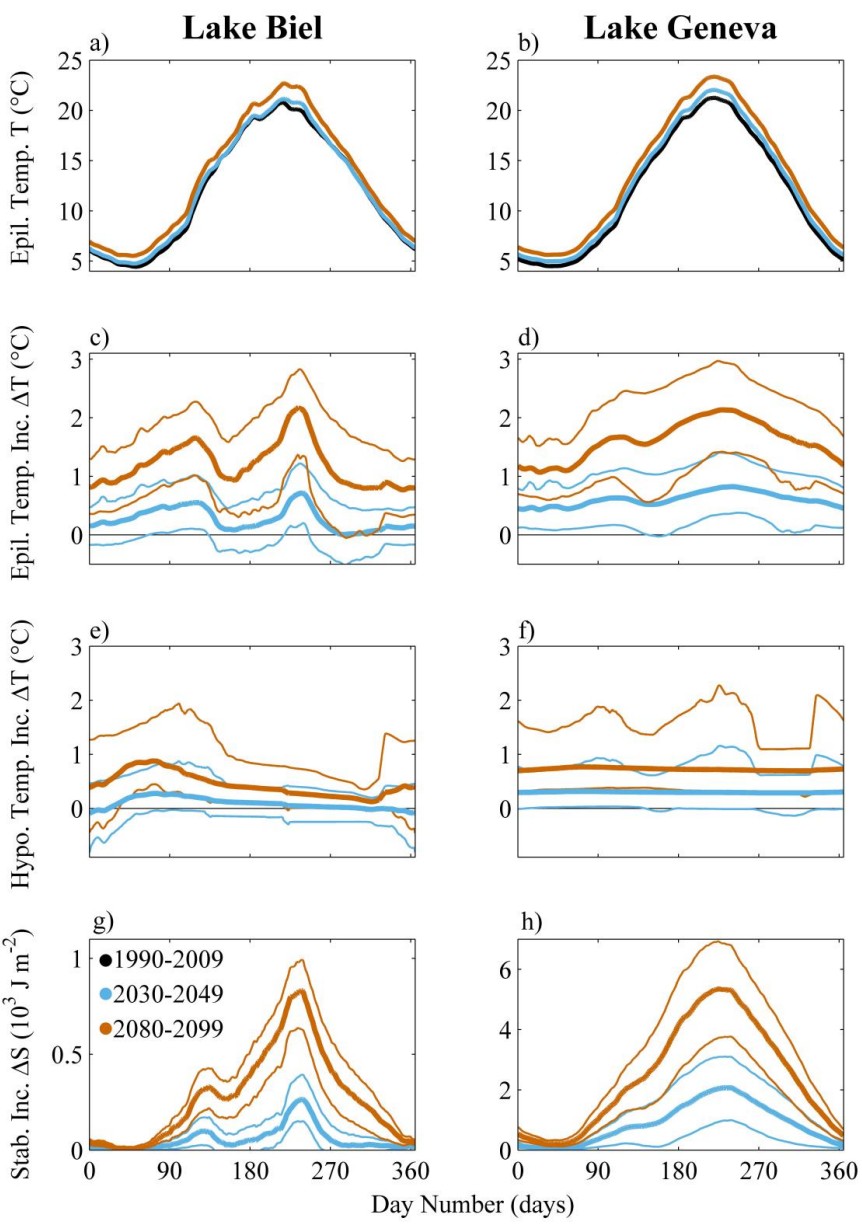

**Figure 7.** Modeled climate impact from scenarios LB3 (LB, left column) and LG3 (LG, right column) displayed
as daily mean (thick lines) and maximum/minimum model values (thin lines) for near-future (blue, 2030-2049)
and far-future (orange, 2080-2099) time periods relative to the reference period (black, 1990-2009).
Anthropogenic MNPP heat input entering LB has been excluded from near-future and far-future time periods but





retained for the reference period. Temperature T (a and b), increase of temperature ΔT in epilimnion (c and d) and
hypolimnion (e and f) as well as increase in stability ΔS (g and h).

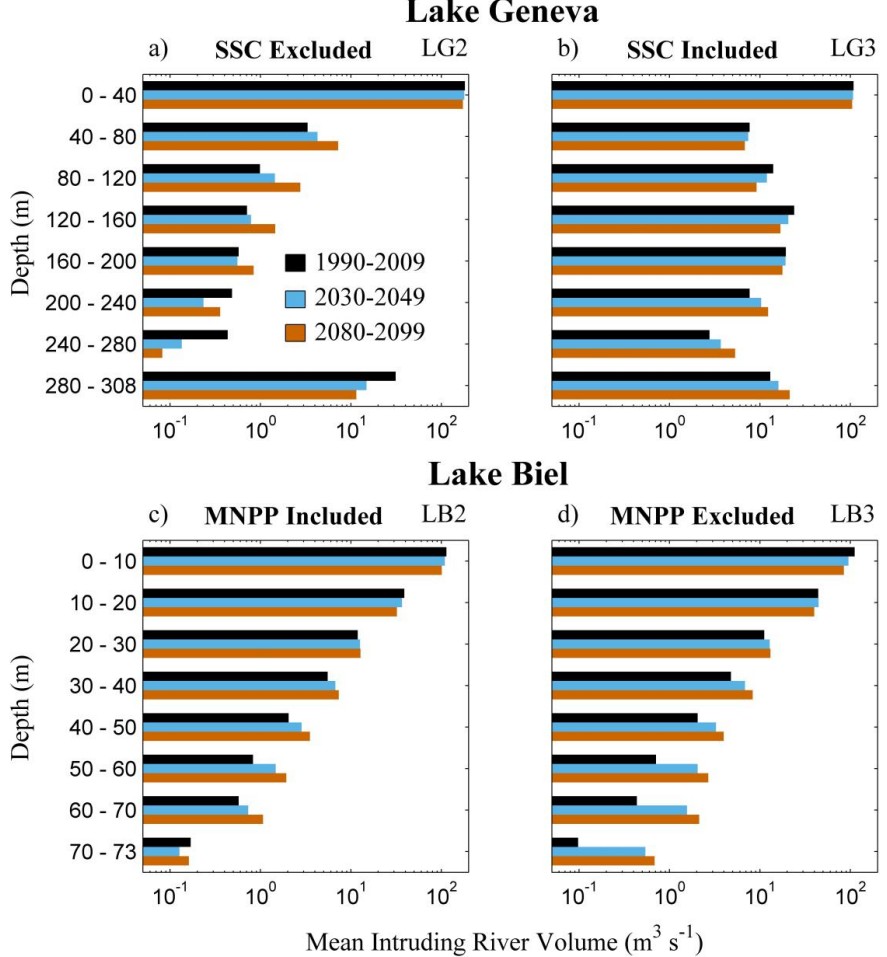


**Figure 8.** Modeled climate impact on mean intruding river volume. Reference (black), near-future (blue) and far-
future (orange) time periods for LG (a to b) and LB (c to d), including (b, c and d) and excluding (a) river borne
SSC and anthropogenic MNPP input included in (c) or excluded from (d) near- and far-future time periods but
retained in the reference period.





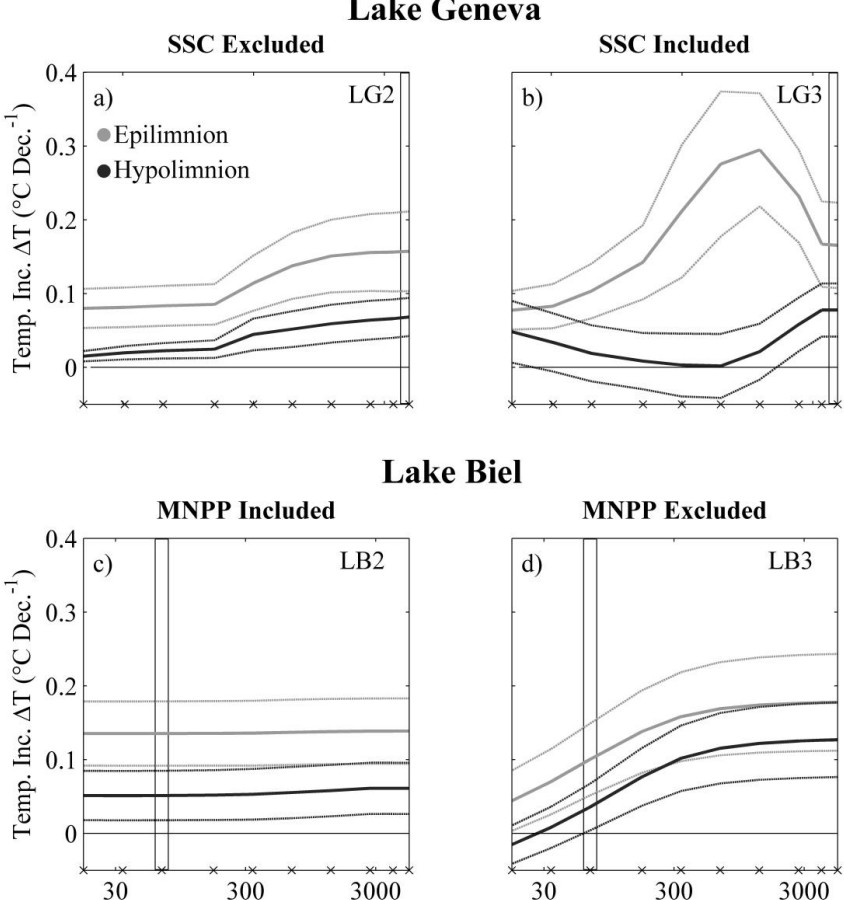


**Figure 9.** Variation in lake hydraulic residence times (changed lake volume) in response to modeled temperature

increase (ΔT) in the epilimnion (grey) and hypolimnion (black) displayed as decadal mean (solid line) and

standard deviation (dotted line) for LG (a and b) and LB (c to f). River borne SSC included (b) and excluded (a,

c and d), MNPP heat input included in (c) and excluded from (d) near-future and far-future time periods but

retained for the reference period. Black x's mark observed lake residence times and black rectangles mark the

unmodified residence times.




**Appendix Figures**

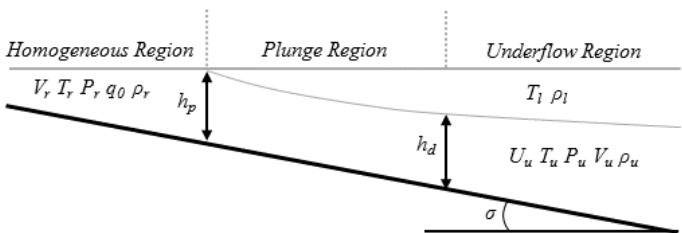


**Figure A1.** Illustration of river intrusion model.

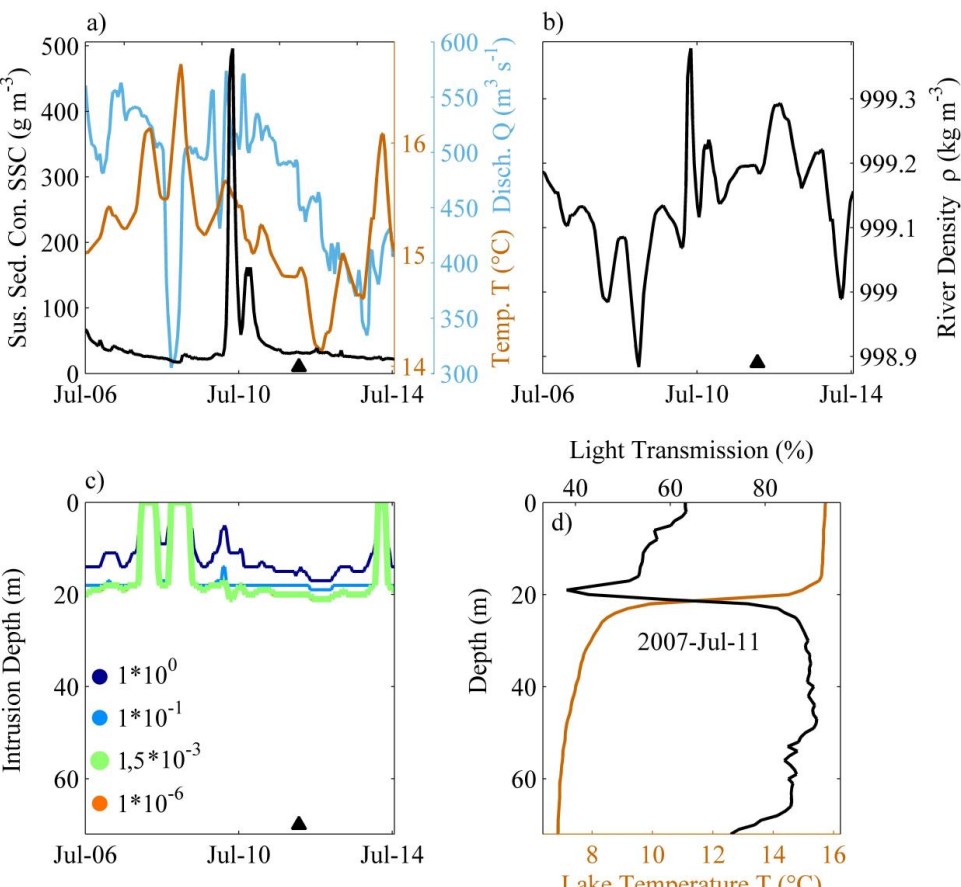





**Figure A2.** River intrusion entrainment sensitivity analysis for the LB/Aare system in July 2007. a) SSC (black),
temperature (T; orange) and river discharge (Q; blue) from Aare station 2085. b) River density at station 2085
obtained from T and SSC in a). c) River intrusion depth calculated from supporting information (S1) with varying
entrainment constant β (Eq. 18); light green denotes the value used in this study. d) Vertical measurements of T
and light transmission in LB for 11th July 2007. ▲ marks the time in a) to c) of the vertical profile in d).

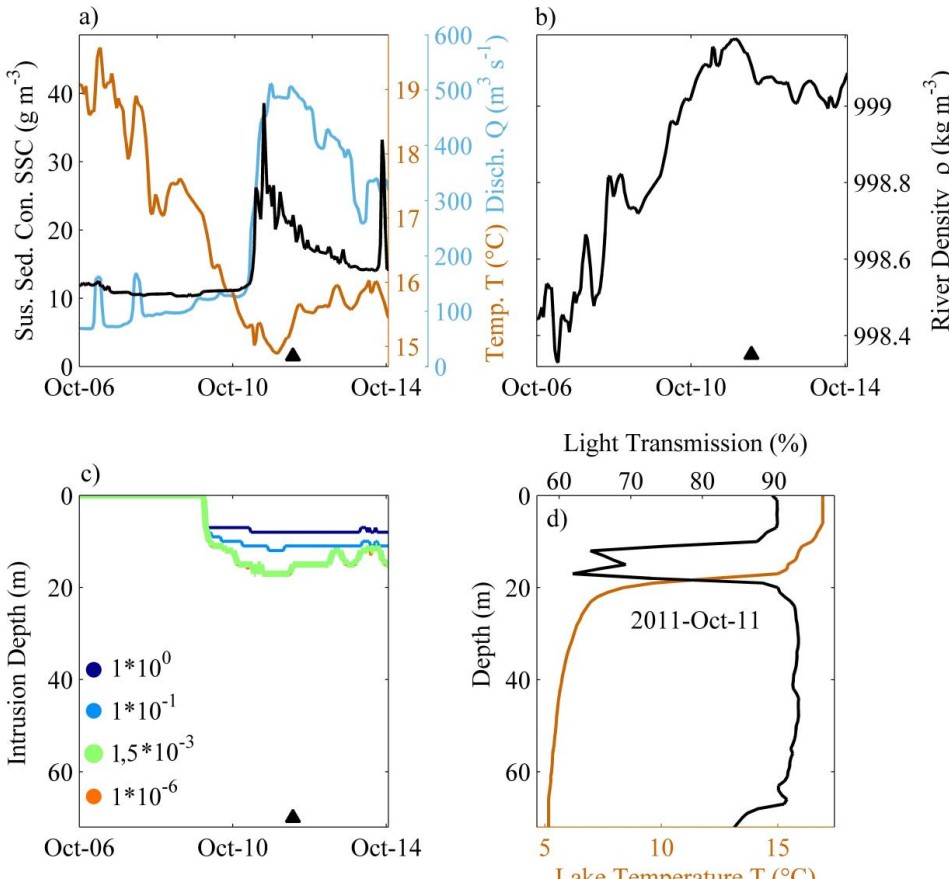


**Figure A3.** River intrusion entrainment sensitivity analysis for the LB/Aare system in October 2011. a) SSC
(black), temperature (T; orange) and river discharge (Q; blue) from Aare station 2085. b) River density at station
2085 obtained from T and SSC in a). c) River intrusion depth calculated from supporting information (S1) with
varying entrainment constant β (Eq. 18); Light green denotes the value used in this study. d) Vertical
measurements of T and light transmission in LB for 11th October 2011. ▲ marks the time in a) to c) of vertical
profile in d).




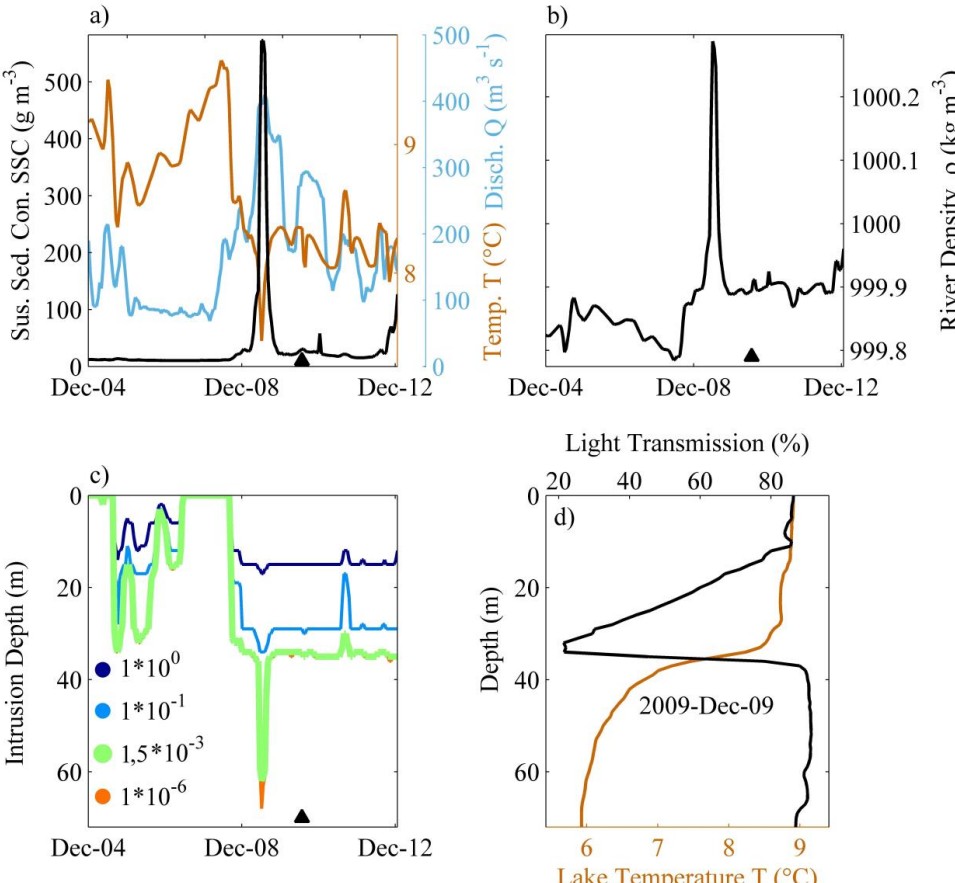

**Figure A4.** River intrusion entrainment sensitivity analysis for the LB/Aare system in December 2009. a) SSC
(black), temperature (T; orange) and river discharge (Q; blue) from Aare station 2085. b) River density at station
2085 obtained from T and SSC in a). c) River intrusion depth calculated from supporting information (S1) with
varying entrainment constant β (Eq. 18); light green denotes the value used in this study. d) Vertical measurements
of T and light transmission in LB for 9th December 2009. ▲ marks the time in a) to c) of vertical profile in d).

840

**Figure B1.** Modeled climate impact on LB excluding river borne SSC. Temperature increase ΔT (a and b) displayed as means (bars) and standard deviations (black lines) in epilimnion (left bar group), metalimnion (middle bar group) and hypolimnion (right bar group); mean intruding river volume (c and d) and mean river





intrusion depth (e and f). MNPP thermal input included (a, c and e) or excluded (b, d and f) in near-future (blue)
and far-future (vermilion) time periods but retained in the reference period (black).

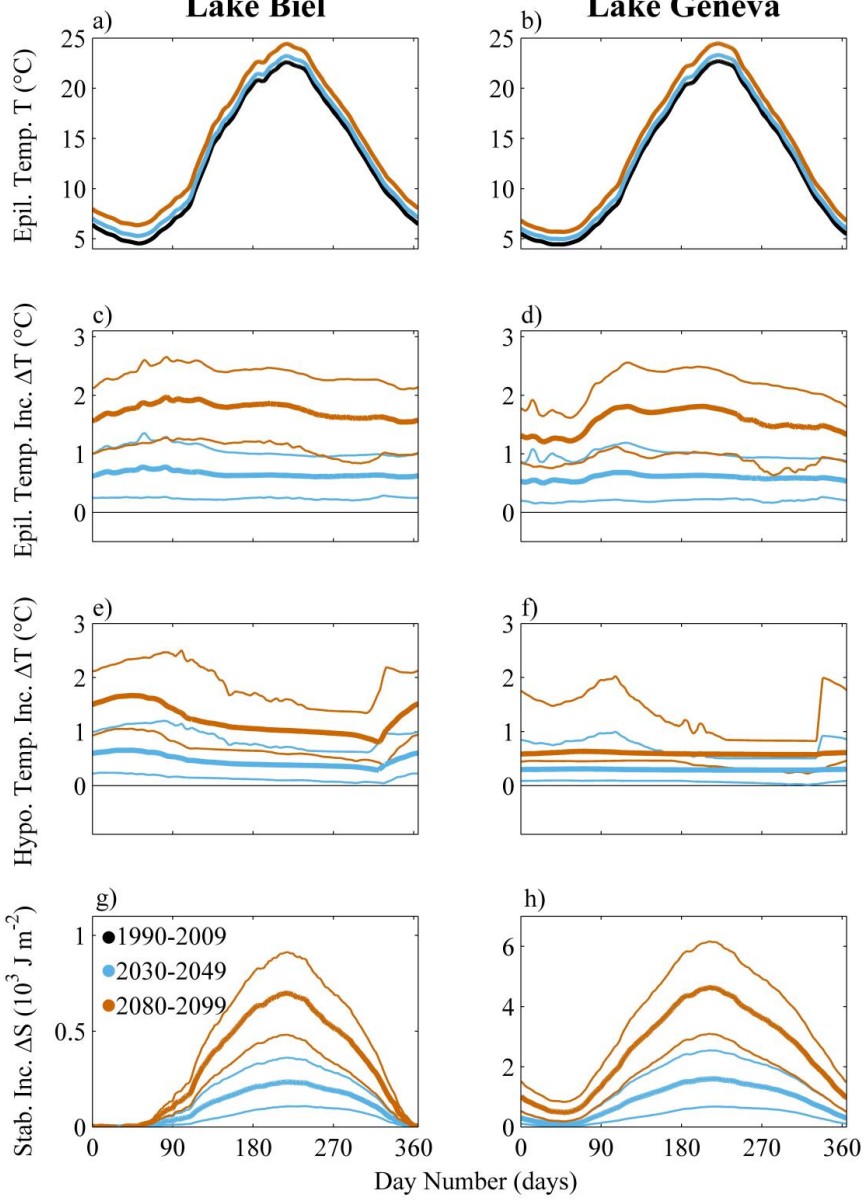




**Figure C1.** Modeled climate impact (river intrusion excluded) on LB (left column, scenario LB1) and LG (right
column, scenario LG1) shown as daily mean (thick lines) and maximum/minimum model values (thin lines) for
near-future (blue, 2030-2049) and far-future (orange, 2080-2099) time periods relative to the reference period
(black, 1990-2009). Temperature T (a and b), temperature increase (ΔT) in the epilimnion (c and d) and
hypolimnion (e and f) as well as increase in stability (ΔS; g and h).

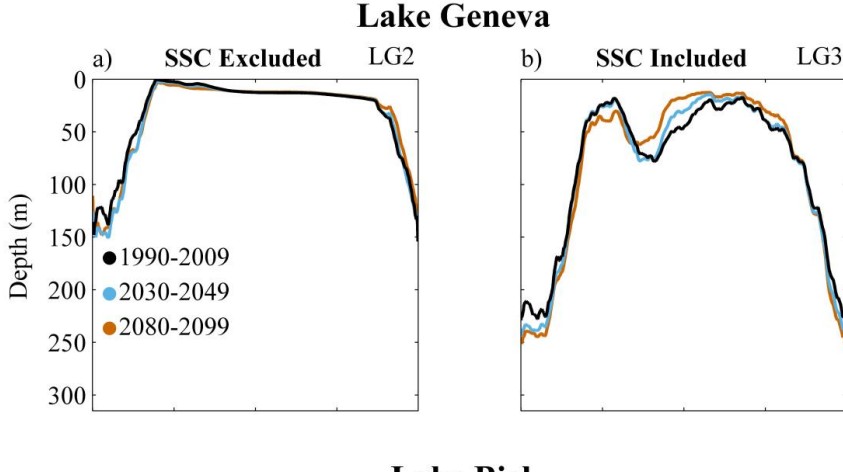

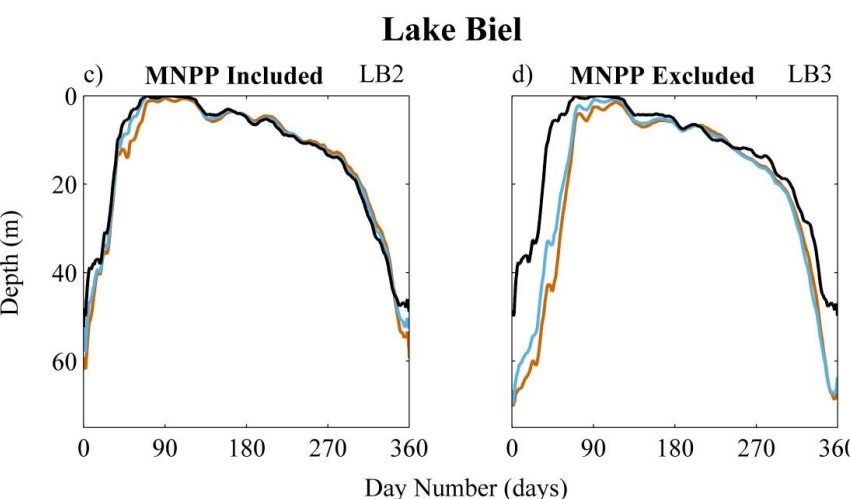


**Figure D1.** Modeled climate impact on mean river intrusion depth. Reference period (black), near-future (blue)
and far-future (orange) time periods for LG (a to b) and LB (c to d) with (b, c and d) and without (a) river borne
SSC and MNPP thermal input included (c) or excluded (d) from near-future and far-future time periods relative
to the reference period.