# Peer review of "Tributaries affect the thermal response of lakes to climate"

_Hydrology and Earth System Sciences, 2017_

## Referee Comment (RC1) · Anonymous Referee #1 · 20 Jul 2017

Specific comments:

1. Section 2.1.(ii) refers to a state-of-the-art river intrusion scheme. The authors provide no comparison to other river intrusion schemes in the literature, and so this claim is difficult to justify, particularly as this type of river intrusion model has been around for 30+ years. Additionally, Appendix A1 (where the intrusion scheme is described) does not identify why this approach was used over other approaches in the literature, or what makes this approach an advance on other approaches in the literature. The authors need to assist the reader here – if you have done the work to show this is the best model, then explain why.

2. Some discussion is required for the SSC model in the Aare River. The inset is missing from Figure 4a (which is present in Figure 4b, and Figure 3a, 3b), presumably

because the r2 of this model isn't great (Table 2). I expect the reason for this is that the SSC in the river is almost always zero (or at least very low), with occasional large spikes, as compared to the Rhone where there is a non-zero SSC for much of the year. So clearly modelling SSC in the Aare is complex – what does this mean for the study? How does this uncertainty propagate to the results that are built upon this SSC model? Following on from this, there is a statement in section 4.3 that the SSC doesn't really even matter for the Aare – so why not include this earlier in the paper, and remove iSSC from consideration in the Aare?

3. Given the nature of the study, and the delicate balances referred to, a comprehensive discussion of uncertainty is required. For example, how do we know that the results shown in Table 5 are signal and not noise? You report small numbers over decadal timescales, and so model results will be sensitive to the parameters selected – how robust are these results to uncertainty in the parameters? There are a number of comments following here that discuss uncertainty.

4. Figure 8b demonstrates that it is the SSC concentration that causes greater volumes of deep water renewal under a warming climate. That is, the SSC overcomes the temperature effects and reverses the trends in Figure 8a. This indicates the model results (and a key part of the papers conclusions) are highly sensitive to the SSC model used. Given that the RMS error is approx. 200 g/m3, what implications does that have on these results?

5. Given that the SSC increases under a climate change scenario, what does this imply for the SSC model? The SSC model only has flow as an input, indicating that the changing flow pattern with climate change is enough to cause higher SSC at just the right time to cause the deeper intrusions?

6. I would like to see the comments in the abstract tied in far more strongly with the material in the manuscript. In particular, the key component of the abstract that will capture the attention of readers is that you expect to see more deep water renewal.

This component of the results and discussion needs to be made far clearer – give it the attention it deserves, so readers can make the link between abstract and text.

Technical comments

7. Line 538 contains a typographic error, I believe the word "dishrag" should be discharge.

8. Figure 6 caption refers to mean and standard deviation – of what? It is not clear.

---

## Referee Comment (RC2) · Anonymous Referee #2 · 30 Jul 2017

This is an interesting paper examining the effect of climate change on alpine lakes. Climate change exerts a dual influence on alpine rivers: changes flow rate and temperature. River temperature and sediment load influence density, and hence the nature of river intrusion into lakes. Future river discharge rates, temperatures and SSC are all predicted in order to assess the future dynamic state of two lakes, Lake Biel (LB) and Lake Geneva (LG), out to the far-future of 2099. The work is interesting and certainly very appropriate for HESS.

There are two parts to the work, the prediction of inflow conditions of the two main inputs to LB and LG, respectively, and secondly the lake dynamics. I believe the assumptions and range of expected behaviors for the first part are well documented, but more discussion is needed for the lake dynamics part.

The paper uses the 1-D Lake model SIMSTRAT. There is very little discussion of SIM-STRAT and, given its key importance in the forecasting, some more details are needed here so that the paper is more self-contained. SIMSTRAT has eight numerical constants, and the assumed values are summarized in Table 3 for each Lake, and there is little explanation of what these constants represent. A key point is these constants are considered fixed, based on best estimates for current conditions. Why will these parameters also valid out to 2000?

Figures 7g and 7h shows increased stability, particularly in the far-future scenario for both LB and LG. This is particularly associated with predicted warming of the epilimnion. So estimating the downward transfer of heat and vertical mixing generally is key to model predictive performance. In future, why can we assume such key quantities, as the downward penetration of radiation into the water column will be unchanged? If the density stratification changes due to climate change, the internal wave climatology will also likely evolve, so can we assume the mixing is the same?

Also lake volumes can evolve, leading to potential changes in residence time as discussed in Figure 9 and text around lines 455 and following. So what is the uncertainly associates with the use of a 1D model like SIMSTRAT in some of these scenarios? The two lakes currently have very different residences times, and LG already has an 11.5 y residence time, so how accurate is a 1-D assumption even now, let alone into the future? These issues need to be clarified in the paper and their influence on the uncertainly of the predictions in the paper.

Please also note the supplement to this comment:
https://www.hydrol-earth-syst-sci-discuss.net/hess-2017-337/hess-2017-337-RC2-supplement.pdf
* * *

---

## Author Comment (AC1) · 18 Aug 2017

**Answers to Reviewer #1**

We thank Reviewer 1 for valuable feedback to this manuscript. Hereunder follow our answers (normal text) to the reviewers' comments (marked in *italic*).

*1. Section 2.1. (ii) refers to a state-of-the-art river intrusion scheme. The authors provide no comparison to other river intrusion schemes in the literature, and so this claim is difficult to justify, particularly as this type of river intrusion model has been around for 30+ years. Additionally, Appendix A1 (where the intrusion scheme is described) does not identify why this approach was used over other approaches in the literature, or what makes this approach an advance on other approaches in the literature. The authors need to assist the reader here – if you have done the work to show this is the best model, then explain why.*

The intrusion scheme was chosen since it enables us to include the effect of steep bathymetry in both lakes. This affects the plume decent speed and thus entrainment and intrusion depth. Additionally, the robustness of the intrusion model outweighs the model simplicity, and limits the uncertainty associated with more complex intrusion schemes including multiple parameters which can be hard to predict in the future. Here we look at the long-term difference, thus errors do not change from reference period to future periods. Also, the 3D movement of the stratified lake water body challenges an intrusion model (as the intrusion condition are changing continuously). As we only model the lake water column in 1D, a sophisticated intrusion model is therefore not an option. Using multiple scenarios (min, mean and max predictions) of both river discharge and air temperature, with corresponding river densities, we cover the likely depth fluctuations of future river intrusions. The aim here was not to use the most sophisticated intrusion model but one which directly translates the density changes in the inflowing rivers to the intrusion depth relative to the lake stratification. As the focus of this paper is to compare different scenarios, the modelled changes between Scenario A and Scenario B is much more representative than the absolute accuracy of a model scenario alone (which is already quite adequate as the figure below shows). We did compare the intrusion scheme used here with a recently published model by *Cortés et al.* [2014], and obtained a worse representation of the river intrusion depth.

This comparison was left out from the original manuscript but will be included in the revised Appendix A1. "State of the art formulation" will be removed from section 2.1 and a short explanation will be given on why a simple model was used. Furthermore, a detailed comparison between modelled intrusion depth and ADCP measured intrusion depth (see figure below) will be included in Appendix A1.

[Figure]

**Figure A5.** River intrusion entrainment sensitivity analysis for the LB/Aare system in July 2014. a) SSC (black), temperature (T; orange) and river discharge (Q; blue) from Aare station 2085. b) Lake temperature at M3 station. c) River intrusion depth calculated as in supporting information (S1) using river/lake density obtained from a) and b) with varying entrainment constant β (coloured, Eq. 18); light blue denotes the value used in this study; intrusion depth (black) calculated with method described in *Cortés et al.* [2014]. d) Current speed obtained from ADCP at M3 station; velocities > 0.15 m s⁻¹ are associated with the passing river plume.

*2. Some discussion is required for the SSC model in the Aare River. The inset is missing from Figure 4a (which is present in Figure 4b, and Figure 3a, 3b), presumably because the r2 of this model isn't great (Table 2). I expect the reason for this is that the SSC in the river is almost always zero (or at least very low), with occasional large spikes, as compared to the Rhone where there is a non-zero SSC for much of the year. So clearly modelling SSC in the Aare is complex – what does this mean for the study? How does this uncertainty propagate to the*

*results that are built upon this SSC model? Following on from this, there is a statement in section 4.3 that the SSC doesn't really even matter for the Aare – so why not include this earlier in the paper, and remove iSSC from consideration in the Aare?*

As reviewer 1 points out, high SSC events in the Aare River are sporadic and not linked to the discharge at station #2085. This results in unsatisfactorily model representation of high SSC events and low $R^2$ values when using the discharge as the determining variable. These high SSC events mainly occur during summer when Lake Biel is stratified. The intruding river plume is thereby generally captured in the metalimnion as the intrusion model and measurements show (appendix figures A2 and A3). For future studies, where this short-term events are important, such as deep oxygen renewal, a better SSC model is required for Aare including representation of local precipitation patterns in the upstream catchment.

These high SSC events occur during a short time period (hours). Here our focus is on the thermal response of the entire system, short-term deep intrusions events are thereby less important than the overall large river discharge patterns. The temporally dominant low SSC in the Aare River over time was successfully reproduced by the model. This affect the intrusion depth only in winter (Figures B1e,f and D1c,d) when stratification is weak and present river discharge is low. However, river discharge is predicted to increase in winter. Furthermore, including SSC leads to consistency in comparison between Lake Biel and Lake Geneva. For these two reasons, we prefer to keep SSC also for the Aare River, although its effect on the intrusion dynamics is rather limited.

*3. Given the nature of the study, and the delicate balances referred to, a comprehensive discussion of uncertainty is required. For example, how do we know that the results shown in Table 5 are signal and not noise? You report small numbers over decadal timescales, and so model results will be sensitive to the parameters selected – how robust are these results to uncertainty in the parameters? There are a number of comments following here that discuss uncertainty.*

Here we keep the natural variability of the hydrological and meteorological systems and apply a seasonal varying climate change signal in temperature and river discharge. Thus, the effect of rapid natural variations in forcing, probably referred to as noise by the reviewer, was kept both in the reference and future periods. We are therefore confident that the change we observe are attributed to our applied climate signal since the "noise levels" were kept from the reference to near-future and far-future.

For the final manuscript, we will add a lake model sensitivity analysis for the important parameters wind, cloud and humidity. These parameters are likely to change in a future climate but has been kept constant here due to the limitations in the current climate model predictions. The analysis will be made from observed historical variability. For the parameter SSC, see also the answers to points 4 and 5.

*4. and 5. Figure 8b demonstrates that it is the SSC concentration that causes greater volumes of deep water renewal under a warming climate. That is, the SSC overcomes the temperature effects and reverses the trends in Figure 8a. This indicates the model results (and a key part of the papers conclusions) are highly sensitive to the SSC model used. Given that the RMS error is approx. 200 g/m3, what implications does that have on these results?*

*Given that the SSC increases under a climate change scenario, what does this imply for the SSC model? The SSC model only has flow as an input, indicating that the changing flow pattern with climate change is enough to cause higher SSC at just the right time to cause the deeper intrusions?*

The reviewer refers to Lake Geneva, where the density change is driven by increased SSC in winter. For Lake Biel, the density change is caused by the power plant decommission. The 200 g/m3 RMSE is obtained from the start of 2013 to the end of 2014. The errors are mainly caused by the difference between model and measurements for extreme high SSC events in summer. In winter, from October to March each year, the RMSE drops to ~88 $g/m^3$. During this period, we observe an increase in the modelled deep water renewal rate in Lake Geneva, at the same time as the performance of the model increase.

SSC in rivers depend on the local land use/soil type, river bed erosion, temporal precipitation patterns/river discharge and glacial cover. How these processes change in the future are hard to predict. However, the main effect controlling SSC is the river discharge, incorporating many of the processes above, here we thus only disregarded land use. The simplicity of the discharge-dependent SSC model (and the river temperature model) is thus its biggest strength. Yet, caution is advised for drainage areas which include sediment trapping lakes/reservoirs, such as the Aare River. The dams constructed in the Rhône catchment are located at high altitude far from Lake Geneva, thereby limiting the sediment trapping effect and enables good results using the SSC model.

As climate changes, the river discharge in winter increases. In the model, as well as in the real world, erosion rates are thus likely to rise. Furthermore, with increased discharge more sediment is carried at slower flow velocities ($<$ threshold discharge $Q_{th}$) thereby increasing the sedimentation of particles in the catchment, which at high discharge events are available for erosion. The density of Rhône River water thus increases at high discharge events due to increased erosion, which can act on a larger deposited sediment mass. This increased river water density/volume in combination with the weak lake stratification in winter result in, (i) increased volume of existing intrusions events past 200 m depth, and (ii) enabling intrusion past 200 m depth by river water previously intruding at a shallower depth. In this study, we keep the natural variability in discharge and only adapt the daily base line. Thus, the frequency of deep penetrating intrusion events is only increased in winter by (ii). Concluding, as rain patterns shift towards winter at the same time as less water is bound in snow and ice, erosion rates increase. This in combination with weak winter stratification strength enables more water to reach deeper layers.

In order to increase the clarity of the manuscript, the following will be added:

To Section 4.1, line 331:

This is caused by two phenomena, (i) amplified river bed erosion linked to increased intensity of high discharge events carrying enhanced volume in the future, and (ii) to increased sedimentation in the river catchment, available for erosion, at slower flow velocities due to additional sediment being carried by increased discharge.

To section 4.2, line 390:

Concluding, as river water density increase in winter the volume of existing intrusions events occurring in the reference period increase. Likewise, high discharge events previously unable to penetrate into the deep is likely to do so in the future.

*6. I would like to see the comments in the abstract tied in far more strongly with the material in the manuscript. In particular, the key component of the abstract that will capture the attention of readers is that you expect to see more deep water renewal. This component of the results and discussion needs to be made far clearer – give it the attention it deserves, so readers can make the link between abstract and text.*

We thank the reviewer for this important note and will adapt the revised manuscript accordingly.

*Technical comments*

*7. Line 538 contains a typographic error, I believe the word "dishrag" should be discharge.*

Correction will be made in manuscript

*8. Figure 6 caption refers to mean and standard deviation – of what? It is not clear.*

This refers to the mean and standard deviation of nine different model runs, combining the lower, median and upper predictions for future air temperature and river discharge. Figure capitation will be updated in revised manuscript to remove this uncertainty.

References

Cortés, A., W. E. Fleenor, M. G. Wells, I. de Vicente, and F. J. Rueda (2014), Pathways of river water to the surface layers of stratified reservoirs, *Limnol. Oceanogr.*, *59*(1), 233–250, doi:10.4319/lo.2014.59.1.0233.

---

## Author Comment (AC2) · 18 Aug 2017

**Answers to Reviewer #2**

We thank Reviewer 2 for valuable feedback to this manuscript. Hereunder follow our answers (normal text) to the reviewers' comments (marked in *italic*).

*This is an interesting paper examining the effect of climate change on alpine lakes. Climate change exerts a dual influence on alpine rivers: changes flow rate and temperature. River temperature and sediment load influence density, and hence the nature of river intrusion into lakes. Future river discharge rates, temperatures and SSC are all predicted in order to assess the future dynamic state of two lakes, Lake Biel (LB) and Lake Geneva (LG), out to the far-future of 2099. The work is interesting and certainly very appropriate for HESS.*

*There are two parts to the work, the prediction of inflow conditions of the two main inputs to LB and LG, respectively, and secondly the lake dynamics. I believe the assumptions and range of expected behaviours for the first part are well documented, but more discussion is needed for the lake dynamics part.*

*The paper uses the 1-D Lake model SIMSTRAT. There is very little discussion of SIMSTRAT and, given its key importance in the forecasting, some more details are needed here so that the paper is more self-contained. SIMSTRAT has eight numerical constants, and the assumed values are summarized in Table 3 for each Lake, and there is little explanation of what these constants represent. A key point is these constants are considered fixed, based on best estimates for current conditions. Why will these parameters also be valid out to 2099?*

The SIMSTRAT model has successfully been used in many different lakes with very different eco-physical boundary conditions (and therefore different calibration coefficients). It is obvious that some of the constants would change with different boundary conditions in the future. However, to affect our model results these changes have to be large and are usually not associated with climate change. For example, if the trophic status would change to highly eutrophic or to ultra-oligotrophic, then the absorption coefficient would change. Likewise, constructions of dams would ultimately alter river discharge patterns and thereby river temperature and SSC content. The most critical phenomenon associated with climate change for our model results, are the glacier retreating rate. Fortunately, as shown by *FOEN* [2012] and stated in the manuscript, this rate is not expected to decrease the glaciers in the catchments used here past ~30% of today's glacier extent. Thus the model constants, calibrated and validated for current conditions, are expected to be useable as long as the systems are not severely structurally altered. As climate change, re-evaluation of the model constants will be required. However, the parameter change is not expected to significantly alter the model results.

We agree with the reviewer that the description of SIMSTRAT can be extended. We propose that the revised manuscript will contain a more detailed section 2.4, describing the main features of SIMSTRAT including clarification of the model parameters and what they represent. However, a detailed description of the model with all equations are available in *Goudsmit et al.* [2002], and will thus not be repeated in this manuscript.

*Figures 7g and 7h shows increased stability, particularly in the far-future scenario for both LB and LG. This is particularly associated with predicted warming of the epilimnion. So estimating the downward transfer of heat and vertical mixing generally is key to model predictive performance. In future, why can we assume such key quantities, as the downward penetration of radiation into the water column will be unchanged? If the density stratification changes due to climate change, the internal wave climatology will also likely evolve, so can we assume the mixing is the same?*

Turbulent mixing (turbulent diffusivity) is dependent on both the kinetic energy input into the system (wind) which is enhancing mixing, and the stabilising effect by heating which reduces mixing. Therefore, the reviewer is correct in that the downward transport of heat will change with altered forcing and stratification. Such changes are however already included in the k-epsilon turbulence closure and therefore included for changing climate forcing.

In this particular application, we only need to consider the increased stabilizing effect due to increased air temperature which limits mixing. In general, parameters which are expected to change in the future and affect stratification and mixing include air temperature, turbidity (light penetration), wind, cloud cover and humidity. The confidence in future prediction of wind, cloud cover and humidity remain still too low [*CH2011*, 2011] and have therefor been kept constant here.

The sensitivity of SIMSTRAT to changed light penetration during climate change in Lake Geneva has already been investigated by *Schwefel et al.* [2016]. Who showed that lower transparency (increased absorption), warms the surface more, strengthens the thermocline and overall cools the deeper layers of the lake. Opposite, increased transparency (weaker absorption) heats the lake surface less and the deep-water more and therefore causes a weaker thermocline.

In the revised manuscript we will include a sensitivity analysis of SIMSTRAT using observed long-term fluctuations of the atmospheric forcing. The sensitivity analyses done by *Schwefel et al.* [2016] will also be discussed.

*Also lake volumes can evolve, leading to potential changes in residence time as discussed in Figure 9 and text around lines 455 and following. So what is the uncertainly associates with the use of a 1D model like SIMSTRAT in some of these scenarios? The two lakes currently have very different residences times, and LG already has an 11.5 y residence time, so how accurate is a 1-D assumption even now, let alone into the future? These issues need to be clarified in the paper and their influence on the uncertainly of the predictions in the paper.*

Here we use the one-dimensional (1D) model SIMSTRAT, thereby horizontal averaging all lake process. The simplicity of the one-dimensional approach is its main strength for long-term (far into the future reaching) climate studies, enabling long temporal scales including natural variability to be modelled under a multitude of different climate scenarios. The performance in Lake Biel of SIMSTRAT has been compared to the state of the art three dimensional (3D)

model Delft-3D by *Råman Vinnå et al.* [2017]. Showing that lake-wide processes could be equally good represented in both 1D and 3D. The difference between these models lays in the representation of local processes. In fact the term uncertainty should not be used in climate research, where future predictions only valid for certain scenarios are considered. Here, we give the possible range of our model results under the A1B emission scenario, as well as our models performance in the past (reference period).

A change in volume has to be extremely large, which is topographically not possible, in order to affect the systems considered here due to the depth of such lakes. The predicted shift in river discharge regime flattens the discharge curve, while maintaining the overall volume entering into both lakes. Assuming no drastic local river altering takes place in the future, a change in volume of both systems would require a geological temporal scale, outside the scope of this study.

References:

CH2011 (2011), *Swiss climate change scenarios CH2011*, published by C2SM, MeteoSwiss, ETH, NCCR Climate, and OcCC, Zurich, Switzerland, 88 pp. ISBN: 978-3-033-03065-7.

Federal Office for the Environment FOEN (publ.) (2012), *Effects of climate change on water resources and waters, Synthesis report on "Climate Change and Hydrology in Switzerland" (CCHydro) project*, Federal Office for the Environment, Bern. Umwelt-Wissen No 1217: 74 S.

Goudsmit, G.-H., H. Burchard, F. Peeters, and A. Wüest (2002), Application of k-ε turbulence models to enclosed basins: The role of internal seiches, *J. Geophys. Res.*, *107*(C12), 3230, doi:10.1029/2001JC000954.

Råman Vinnå, L., A. Wüest, and D. Bouffard (2017), Physical effects of thermal pollution in lakes, *Water Resour. Res.*, *53*(5), 3968–3987, doi:10.1002/2016WR019686.

Schwefel, R., A. Gaudard, A. Wüest, and D. Bouffard (2016), Effects of climate change on deepwater oxygen and winter mixing in a deep lake (Lake Geneva): Comparing observational findings and modeling, *Water Resour. Res.*, *52*(11), 8811–8826, doi:10.1002/2016WR019194.